# The effects of skill-based health education—A randomised-controlled intervention in primary schools in rural Bangladesh

Makiko Omura[1]*, Mohini Venkatesh[2], Ikhtiar Khandaker[3¤a], Ataur Rahman[3¤b]

1 Department of Economics, Faculty of Economics, Meiji Gakuin University, Tokyo, Japan, 2 Department of Education and Child Development, International Programs, Save the Children Federation, Inc., Fairfield, Connecticut, United States of America, 3 School Health and Nutrition Program, Save the Children Bangladesh, Dhaka, Bangladesh

¤a Current address: Care Bangladesh, Mohakhali, Dhaka, Bangladesh
¤b Current address: Maternal & Child Washing (MCW) Project, Result for Development, Dhaka, Bangladesh
* makiko@eco.meijigakuin.ac.jp

## Abstract

This paper evaluates the impact of skill-based health education (SBHE) on children's hygiene practices and health in rural Bangladesh. Over one year, SBHE was delivered weekly to primary schools through a randomised-controlled trial (RCT) by locally recruited trained para-teachers. The SBHE and soap provision interventions, provided in a cross-cutting manner, were randomly assigned to 180 schools stratified by two school types. Data were collected at both the school and child levels, involving 40 students in grades 1–4 at baseline and expanding to 50 students in grades 1−5 at endline, ten students per grade. The study tracked the same pupils, supplemented by additional and replacement students, resulting in effective sample sizes of 7,192 at baseline and 8,992 at endline. The results indicate that SBHE significantly improved hygiene practices; the average treatment effects on overall hygiene practices was 0.22 SD [0.14–0.31] (p < 0.001), and those on health/hygiene knowledge was 0.44 SD [0.33–0.55] (p < 0.001). While improvements in cold-related symptoms among the SBHE school children were marginally significant (−0.05 SD [−0.10, −0.01], p = 0.09), the overall trends indicated general improvement in health as well as healthy behaviours across all schools, irrespective of treatment status. Additional analysis incorporating inter-school spillover effects provided robust evidence of beneficial healthy practice externalities, extending beyond treatment school children. A cross-cutting soap provision treatment, although implemented imperfectly, did not show any standalone positive health-related effects. Nonetheless, our cost-effectiveness analysis indicated the economic viability of SBHE, particularly when accounting for spillover effects. This study is registered in the AEA RCT registry (No.0004265) and the ISRCTN registry (No.18002856).

**Data availability statement:** Data cannot be shared publicly because the dataset contains sensitive information on minors' health and household conditions, and explicit participant consent for public data sharing was not obtained. Data are available from the corresponding author for researchers who meet the criteria for access to confidential data, subject to ethical approval and a collaborative research agreement.

**Funding:** This study was enabled by the research grants from the Japan Society for the Promotion of Science—Japanese Grant-in-Aid for Scientific Research (No. 23402033) and the Nomura Foundation, and by in-kind contribution and collaboration with Save the Children Federation, Inc. (originally Save the Children, USA). The funders had no role in the study design, data collection and analysis, decision to publish, or preparation of the manuscript.

**Competing interests:** The authors have declared that no competing interests exist.

## Introduction

### General background and study objectives

Global efforts to enhance child health have increasingly emphasised the importance of environmental hygiene and personal health practices, especially in low-resource settings where infectious diseases remain prevalent. Despite substantial progress in public health initiatives, rural areas in developing countries often lag behind in achieving sustainable health improvements. In Bangladesh, notwithstanding concerted efforts by both the government and international bodies, persistent health issues such as diarrhoea, micronutrient deficiencies, and respiratory infections underline the continuing challenges in school-aged populations [1–3], exacerbated by insufficient hygiene facilities [4]. This highlights the ongoing need for robust public health strategies and interventions to secure the well-being of Bangladeshi children.

While the major interventions for child health have rested on health commodity provisions, such as drugs, vaccines and hygiene infrastructure, the significance of school health education has gain much attention recently [5–8]. Skill-based health education (SBHE) has emerged as a promising approach to instil effective hygiene practices among school children by focusing on skill acquisition and habit development beyond simple knowledge transmission [9,10]. The effects of acquired skill and habit can be sustained without perpetual external inputs, and by promoting long-lasting healthy behaviours, such as regular handwashing and proper sanitation use, it is expected to prevent common illnesses [11,12].

This study seeks to evaluate the effectiveness and cost-effectiveness of school-based SBHE in primary schools in rural Bangladesh. By adopting a cluster randomised-controlled trial (RCT), this research examines whether SBHE can lead to improved hygiene practices and overall health status among children. Such evidence is essential for scaling up effective interventions and guiding public health strategies.

### The relevance of SBHE in Bangladesh

Maintaining a healthy school environment and habit implies better hygiene practices, yet many developing countries face challenges beyond limited budget, such as inadequate practices and management, and resulting short lifetime of hygiene infrastructure. In Bangladeshi schools, the already poorly available infrastructural resources are poorly managed and get underutilised—even newly built latrines become unusable and discarded after several years. Latrines are often locked and/or in bad conditions, and soaps are unavailable or available only at the teachers' room upon request. A case study in primary schools in Niger [13] also highlights this kind of problems. While accessible and adequate hygiene infrastructure is essential for healthy practices [14], changing default behaviour towards healthy and hygienic through SBHE would not only be less costly than building a new infrastructure, but also be realisable in a shorter time, with life-long effects. A cluster-RCT focused on enhancing the upkeep of shared toilets in Dhaka's urban slums demonstrated that promoting behaviour change along with the provision of low-cost simple goods significantly improved maintenance outcomes [15].

### Academic context and rational

Despite widespread recognition of its importance [5], school health education has seldom been rigorously evaluated, particularly in primary education settings [16]. Past studies often reveal that direct provision of drugs and/or supplements, generally cheaper and quicker, typically overshadow health education intervention in immediate effectiveness [17–19]. Nonetheless, evidence from specific areas like oral hygiene [6,7,10] and handwashing [12,20] demonstrates that educational intervention can yield substantive behavioural benefits. The SBHE differs from typical lecture-based health education as it focuses on the context-relevant knowledge, attitudes and skills development in children through interactive learning methods. A quasi-RCT in Egypt and RCT studies in Bangladesh, incorporating topic-specific SBHE, provide positive behavioural and health outcomes [11,20,21]. And yet, assessments of the cost-effectiveness of health education intervention remain sparse [18,20]. This study leverages a robust methodological framework to examine both the impact and the cost-effectiveness of SBHE with a wider topical coverage, aiming to provide fresh insights into the role of school health education in public health.

### Why focus on school children?

Schools offer a strategic platform for instilling healthy habits among large groups of children, inducing economies of scale and peer effects. This study targets school-aged children, a group often overlooked in health interventions, expecting them to be particularly amenable to changes in habits. By incorporating regular SBHE sessions into the school curriculum, the intervention promotes healthy habits through repeated learning-by-doing together with classmates. The importance of repeated learning is attested by [7]. While a child as an individual may feel it cumbersome in the beginning to wash hands, wear shoes, water the latrine before defecation, dispose waste in a waste box, or clean their classroom, etc., they are induced to repeat these actions which can become habitual. Beneficial peer-effects are also expected among the schoolmates [22–27]. Despite habit and behavioural inertia, research suggests that simple, repetitive health actions can become habitual within months [28], emphasising the potential of SBHE to foster lasting health improvements that extend beyond the individual to influence family and community health practices.

The rest of the paper describes the project and intervention design, SBHE strategies, sample selection, statistical analysis in the method section, followed by the result section encompassing the baseline descriptive statistics, estimation results, additional externality/spillover considerations and cost-effectiveness analysis. We then discuss our study findings in terms of their effectiveness and limitations, also in comparison to the existing literature, methodological refection, and the broader implications for public health policy. In the conclusion, we synthesise our findings and propose future research directions.

## Methods

### Project and intervention design

**Location.**  The project was conducted in Jhenaidah district (*zila*), which was a target district of Save the Children (SC)'s PROTEEVA (Promoting Talent Through Early Education) project. However, our intervention specifically took place in two non-PROTEEVA sub-districts (*upazila*), Moheshpur and Kodchandpur, spanning areas of 21.16km² and 20.16km², respectively. These neighbouring sub-districts were chosen due to their need and logistical convenience, given the existing SC offices and local NGO partnerships in the region. To prevent any crossover effects, our project did not involve any PROTEEVA-targeted schools/communities.

**Approvals and registrations.**  The project was approved and supported by the Bangladesh Ministry of Education, Directorate of Primary and Mass Education in Dhaka, the Directorate General of Health Services, and the District Primary Education Officer in Jhenaidah, *Upazila* Education Officer, the schools in the Moheshpur and Kodchandpur *upazilas*. The study obtained an ethical approval by the Research Integrity Review Board of the principal investigator cum lead author's

institute, Meiji Gakuin University. The field experiment was registered with the American Economic Association's registry for randomised controlled trials (AEARCTR-0004265) and the ISRCTN registry (ISRCTN-18002856). The registrations were done retrospectively given the fact that AEARCTR was not yet established at the time of initial participant recruitment, and that health education intervention was not considered to be a clinical trial by the authors at the time of initial participant recruitment concerning the ISRCTN registry. We adhered to the Consolidated Standards of Reporting Trials guidelines for reporting results of RCTs (see Supporting Information S1 File for CONSORT checklist in S1 Zip [29]).

**Randomisation and intervention description.**  The project applied a treatment-control pre-post evaluation based on a cross-cutting randomisation design of SBHE (HE) and a soap-provision (SP) intervention. The unit of intervention was school, and 180 randomly chosen schools out of total of 204 primary schools were stratified according to the main school type—government primary school (GPS) and registered non-government primary school (RNGPS). The number of schools was the maximum feasible number given the budget and logistical constraints. A cross-cutting HE-SP treatments were then randomly assigned to 180 schools, stratified by two school types. Thus, four treatment groups (HE, SP, HESP, and control) with 45 school each were randomly chosen. Treatment randomisation ensured statistical nondifference between treatment and control schools at school-level variables. RNGPSs, despite their names, were actually government-funded, although they received only a fraction of government funds received by GPSs. Given that RNGPSs were disadvantaged in terms of resources and school infrastructure, it was important to stratify by the school type.

Half of the HE-treatment and half of the control schools were assigned to an additional SP-treatment. The SP-treatment was intended to assess whether goods provision per se or interactively with HE could promote healthy behaviour, in this case, handwashing. For SP-treatment, no health advice was provided but each target school and its randomly chosen pupils received six and three small soap bars per month, respectively. The SP-treatment had another role in mitigating possible Hawthorne effects, especially given our primary objective of discerning the impact of SBHE. Hawthorne and John Henry effects are changes in the behaviour of HE-treatment and control groups, respectively, due to the sheer fact of evaluation taking place. In order to mitigate John Henry effects among the pure controls, the project provided game boards to the control schools, which did not affect the measurability of our intervention. Another way to amend these effects would be to collect data for some time, especially after the completion of formal evaluation, but that was outside the scope of this project.

## Samples selection

**School and child selection.**  Among the 204 primary schools in the target sub-districts of Moheshpur and Kodchandpur in Jhenaidah district, comprising 103 GPSs and 101 RNGPSs, 180 were randomly selected, stratified by school type, using Excel random classification formula. The process was repeated until statistical nondifference of baseline school characteristics between the groups was ensured. While the intervention was conducted at school level, data were collected at both school and child levels. Children surveyed were chosen randomly using seat placement based on the pre-determined randomly selected seat numbers prescribed by the principal investigator cum lead author. Surveyors were masked about the treatment status in both baseline and endline surveys. The treatment assignment was done after the baseline data collection, thus all participants were masked about the treatment at the time of baseline survey. Moreover, the treatment assignment was done after the beginning of academic year in Bangladesh, thus the treatment status should not have affected the choice of school by the children and their families.

## Sample size calculation

Observations were made for 180 schools at both baseline and endline, and for 7,200 and 9,000 pupils for the baseline and endline surveys, respectively. The sample size calculation at the time of the baseline survey was based on the expected improvement of 0.15 standardised effect size in child health-related indicators, detected with 80% power and

5% significance level, assuming the school-level covariates to explain 2.5% of the variance. The standardised effect size of 0.15 was a conservative estimate yet deemed to be a reasonable degree of improvement. The sample size calculation utilised the intracluster correlation of 0.058, which was derived from SC's baseline survey data in a different area of Bangladesh in which SC implemented a school health and nutrition project. A sample size of 34 students per schools was deemed sufficient. However, to account for partial compliance and/or attrition, which was assumed to be 20%, we collected data for 10 students, five of each sex, in each grade 1–4, totalling 40 students for the baseline and 50 students in grades 1–5 per school for the endline.

## Data collection

Bangladeshi primary schools are for grades 1–5. Because the sample targets were those who were in primary school during the intervention period, the baseline survey prior to the intervention was conducted for grades 1–4 who would be in grades 2–5 at the time of intervention, and grades 3–6 at the time of the endline survey unless they repeated or dropped out of the class. Additionally, we collected data from ten grade 1or 2 pupils per school in the endline survey, who would have been in grade 1 at the time of the intervention. The endline data were thus collected from grades 1–6, where grade 1 or 2 signified those who had been pre-primary at baseline, and grade 6 students had already graduated from primary school at endline. Children surveyed at baseline were also surveyed at endline, with replacement children of the same sex and same baseline class collected for those who had attritted.

Data collection was done by the Dhaka based survey institute SURCH who received intensive training on the questionnaires, measurement and interview methods, as well as the subject random selection method by the lead author. They also conducted a pilot survey with the authors. For school data, interviews were conducted to headteachers, and observational data were collected with photographs. For child data, interviews and observational data were collected. All interviews and data collection used structured questionnaires. Prior to the study commencement, participation agreement was obtained from the District Primary Education Officer and school headteachers in the meeting. Parents/guardians were briefed on the project details and the possibility of their children participating in the survey conducted at school. They were then asked to provide their consent. At the time of the child survey, written consent was obtained from them using the child assent form. Surveyors were available to assist in completing the forms for the participants when necessary. Among those randomly selected children, some were excluded from the survey if either they or their parents or guardians did not provide consent.

## Skill-based health education strategies

SBHE, while gaining increasing attentions, have often been considered expensive, labour intensive, and requiring long-term involvement, which many practitioners found unsustainable and difficult to replicate to scale, while their impact was hard to discern. The project was thus designed with the following key features: (1) weekly health education sessions using participatory learning methods to promote habit-formation; (2) use of para-teachers instead of primary school teachers as the SBHE facilitators, given their cost-effectiveness and the incentive problems expected among the incumbent teachers; and (3) the use of a mobile projector to assist pedagogy and skill-learning, motivating both pupils and teachers. The device could be easily charged in advance; this was important given Bangladesh's unreliable power supply.

The health education session consisted of 26 modules on seven topics as shown in Table 1. The educational contents utilised many images and videos relevant to the Bangladesh context, which were made appropriate for primary school pupils, as well as learning-by-doing sessions, for handwashing, brushing teeth, saline making, latrine usage, and latrine cleaning, aiding children to acquire health-related knowledge, attitude, practice and behaviour (KAPB). Project staff also recorded various videos, including one featuring a smartly dressed male staff member demonstrating the correct method of cleaning a latrine. The use of such materials aimed to provide practical guidance and to help diminish or eliminate biases against 'dirty

**Table 1. 26 Skill-based Health Education Modules.**

| SBHE modules (26) | Contents |
|---|---|
| 1. personal hygiene (3) | handwashing; hair and nail trimming; bathing, cloth-cleanliness; eye care; teeth-brushing; tooth decay |
| 2. sanitation (3) | keeping premises clean and waste management; latrine cleaning, sanitary latrine use |
| 3. safe water (3) | safe and unsafe water; water purification and preservation; arsenic problem and prevention |
| 4. common illness (6) | fever; cold and cough; diarrhoea cause; diarrhoea prevention and treatment; causes and demerits of worms; deworming |
| 5. nutrition (5) | balanced diet; symptoms of malnutrition; vitamin A; iodine; iron |
| 6. first aid (3) | burns; cuts; fractures |
| 7. injury prevention (3) | burns; drowning; road accidents |

jobs' of latrine cleaning. Each week, a one-hour SBHE session was conducted during the time slot allocated for Physical Education (PE) for each grade. Typically, PE periods were used as free time for children to play. If playing around is assumed to have positive health effects, which may well be the case, the estimated HE effects on health may be biased toward zero.

## Para-teachers

Para-teachers conducted five classes daily at a single school and rotated among five different schools throughout the week, resulting in a total of 25 hours of SBHE sessions each week. They were all recruited locally who were either recent college graduate or previous NGO worker looking for jobs. Some of them also had previous experiences in the health/education field. There were several reasons for recruiting them. Firstly, incumbent teachers were already overwhelmed with their existing curriculum, and similar SC projects had faced significant moral hazards when employing these teachers, despite offering extra payment. Meanwhile, the para-teacher strategy proved to be successful in another SC project in Meherpur district of Bangladesh; some schools that had not been supported by SC started to recruit para-teachers with their own initiatives and funds. Also, an RCT study promoting better school climate/environment showed that a new, low-cost lay counsellor produced substantially beneficial effects while a regular teacher produced no effect [30]. Secondly, younger job seekers would be more flexible and eager to learn and teach the SBHE contents that required the application of skill-based teaching methods and use of a mobile projector. Indeed, para-teachers considered that going through the SBHE training and practices improved their capacity and skills as prospective teachers. Thirdly, it was considered sustainable cost-wise even after the project, as the cost per para-teacher was 5,000 BDT per month, approximately 67 USD, based on the actual market exchange rate at that time, at 1 USD = 74.66 BDT, which was marginally lower than the minimum salary of 5,900 BDT for a regular GPS assistant teacher, yet manageable for the hiring schools, costing them only 1,000 BDT or 13.4 USD per school. Their honorarium was also equivalent to what was being paid by the Meherpur schools stated above, thus considered sustainable even after the project completion. The para-teachers received 40+hours of intensive participatory-training in SBHE by BRAC health education specialists which enabled them to deliver specialised contents.

To date, no comprehensive school health education is offered for primary schools in Bangladesh. The latest health policy states the objective to train at least one teacher per school on health issues [3], however, this kind of arrangement would unlikely be successful for the reasons stated above.

## Project implementation

The intervention was executed for over 12 months from 1 March 2012–25 March 2013, preceded by a four-month period of baseline data collection for which recruitment occurred from 13 October–29 November 2011, mostly during cooler winter season. The endline data collection was conducted for five months after the project's completion, during 7 April–6

July 2013, mostly during hot and humid summer, complemented by further follow-up and data verification processes. The planned intervention period was from January to November 2012, based on the academic year in Bangladesh. However, the start of the implementation was shifted back for a few months due to a procedural delay. The project profile depicting the participant flow with randomisation design is provided in Fig 1.

All HE-treatment schools were delivered all 26 modules on seven SBHE topics by the para-teachers. They also guided a cleaning rota of classroom and latrines by pupils as a part of SBHE, adopting from the Japanese system [31]. There were a few extra weeks as well as several school health events in which skill-trainings for the first two topics and illness treatment, such as ORS making, were repeated. As part of the intervention, the treatment group pupils were encouraged to measure their weight and height occasionally, which was expected to increase their health awareness. The SP intervention was not implemented as planned due to delayed procurement of soap bars

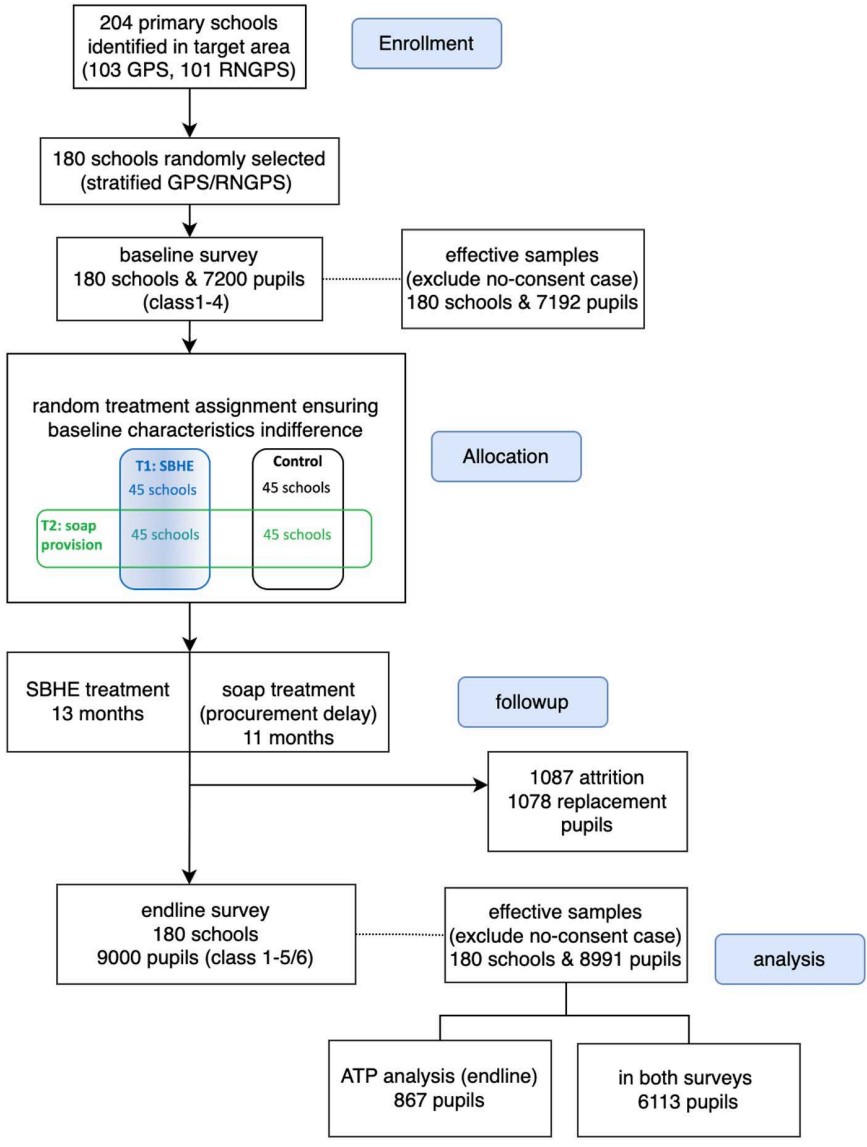

**Fig 1. CONSORT flow diagram: study profile with participant flow and the randomisation scheme.**

for a few months, resulting in their irregular distribution. The SP schools were not monitored in terms of how they used the distributed soap bars. Of these, 12 out of 45 SP schools and 8 out of 45 HESP schools reported not having received the soap bars, although this was likely due to recording errors at the school level, since the target pupils/ households of those schools reported having received the soap bars. While the comparison of HE-treatment and SP-treatment schools was anticipated to provide insight into the direct effect of SBHE, independent of Hawthorn effects, as well as to discern the effect of mere goods provision on promoting healthy behaviour, the imperfect implementation of the SP-treatment raised certain concerns about the validity of this attempt. Thus, the cross-cutting intervention results are mainly contained in the Supporting Information, and the statistical analysis results below mainly focus on the HE-treatment effect.

A refresher workshop was conducted in the midway for their experience-sharing, feed-back and suggestions by the para-teachers and project staff. Some feedback was: difficulty in teaching micro-nutrients; need of additional materials to explain vitamin deficiency; need of shades/curtains for the window to effectively use the projector; problems with locked/ unfunctional latrines; use of projector being beneficial; students liking cartoons; children asking mothers to wash hands properly; children asking mothers to cut their nails; making toilet brush with palm tree leaves.

### Child attrition

Between the baseline and the endline, the study experienced a 15% attrition (1,080 out of 7,192 children), falling within the anticipated 20% attrition outlined in our sample size calculation. Specifically, both HE-treatment and HE-control groups saw attrition rates of 15.0%. While our focus is on HE-treatment effects, attrition rates were 13.7% and 16.3% for SP-treatment and SP-control groups, respectively, and 15.5%, 12.9%, 14.6%, and 17.1% for the HE-only, SP-only, HESP-treatment, and control schools, respectively. This could suggest that soap provision, though imperfectly implemented, might have affected a higher retention rate in SP-schools.

Most of the children in grade 5 during the intervention, thus grade 4 at baseline, had already finished primary school at the time of the endline survey, and their data were collected at nearby high schools, while household follow-ups for those who did not go on to neighbouring high schools proved to be mostly unsuccessful. For students who were in grade 4 at baseline and progressed to grade 5 during the intervention, endline data was gathered from 79.5% of them. Among these, 89.3% had transitioned to nearby high schools, while the remainder repeated the grade in primary school. The school-wide class repetition rate at endline among our sample children amounted to 20.2%, of which 91.3% were single repetition. The repetition figure among our sample was much higher than official school figures of 9.4% on average (see [32,33] for the analysis on schooling). Attritted children mostly had gone to other regions. We collected replacement data for 15% or 1,078 pupils for those who attritted at endline. The replacements were of the same sex and the same baseline class. With the addition of those who were in grade 1 at the time of intervention, the total effective sample size became 8,991 at endline compared to 7,192 at baseline (see Fig 1).

At the child level, there were some systematic differences between attritors and non-attritors suggested by a simple t-test of baseline characteristics. More male, older, and unhealthy pupils—in particular, with more cold-related symptoms in two weeks, more diarrhoea symptoms, and more incidences of fatigue and appetite loss—attritted, although no statistical difference was seen between healthy behaviour and practices. The reasons for attrition could not be retrieved. Regressing attrition on the baseline variables, each interacted with HE treatment, HE-schools had higher rate of attritors experiencing diarrhoea, while HE-control schools had higher rate of attritors experiencing stomachache. No other statistically significant difference was observed between HE-treatment and HE-control schools. This suggested that attrition was unlikely to introduce a clear directional treatment bias. Thus, if attrition has affected the evaluation outcomes, it would likely be a downward bias rather than an upward bias. This would be especially true if the treatment exerted a positive impact, particularly on less healthy and/or older pupils.

## Statistical analysis

In this study, we primarily assess the impact of HE intervention on health and health KAPB at the child level (see [32,33] for the school-level analysis). Since the intervention was done at the school level, only intent-to-treat (ITT) effects can be measured at the child level. At each level, the effects of treatment are measured for family-wise summary indices in which the outcome variables $y$ are categorised into families of related variables, as explained below. The family-wise mean-standardised average treatment effects are estimated through seemingly unrelated regressions (SUR) that allow contemporaneous errors to be correlated [34,35]. This analysis accounts for possible cherry-picking caused by the increased likelihood of finding significant results simply due to conducting many regressions [36,37]. We provide additional estimation results for single outcomes in the Supporting Information. Note that the fact that many of the outcome variables $y$ being binary does not pose any particular challenge in obtaining the average causal effect of a random treatment, as the average treatment effect (ATE) exhibits differences in probabilities of $y = 1$ [38].

Given the fact that 15% of children in the endline survey were replacement of those attritted, we conduct estimations also for the subset of children who were in both baseline and endline surveys. All statistical analysis was conducted using STATA 18.

## Outcomes

This study analyses six primary outcome families concerning health-related KAPB, with two auxiliary outcome families which contain endline-only observations, and three secondary health outcome families. The primary child-level outcomes concern health-related KAPB change, measured through pre-post questionnaires, measurements, and observations. Health-related KAPB outcome families are: (P1) *handwashing practice*; (P2) *dentalcare practice*; (P3) *other hygiene practice*; (P4) *hand cleanliness*; (P4E) *hand cleanliness + adenosine triphosphate (ATP)*; (P5) *nutrition practice*; (P6) *health/hygiene knowledge*; (P6E) *health/hygiene knowledge + extra*.

The *handwashing practice* family (P1) consists of four variables, *handwashing habit index* as explained below, *frequency of handwashing with soap on each occasion* (0 = no; 1 = yes), noted also as (0–1), *washing under running water* (0 = never; 1 = sometimes; 2 = always), and *correct handwashing procedure* (0–8). The *handwashing habit index* reflects both handwashing frequency (0 = no; 1 = sometimes; 2 = always) and materials used on each occasion—before eating, after defecation, and after playing. Precisely, it is the summation of three indices for each occasion, scored according to the handwashing frequency with different material combinations (0 = do not wash; 1 = sometime wash with mud (no soap/water/ash); 2 = always wash with mud (no soap/water/ash); 3 = sometime wash with ash (no soap/water); 4 = always wash with ash (no soap/water); 5 = sometime wash with water (no soap); 6 = always wash with water (no soap); 7 = sometime wash with soap; 8 = always wash with soap). While the usage of soap is considered more hygienic, rubbing hands with ash/mud/soil is considered to be effective in removing faecal coliforms, although the risk of contamination by microbial pathogens, helminths, and chemical toxins is present [39,40]. In rural Bangladesh, washing hands with only water before eating/cooking, or with ash/mud/soil after defecation was widely seen. While these practices were self-reported and not easily measurable, surveyors also assessed the thoroughness of handwashing by evaluating the number of correct steps demonstrated by children.

Self-reported *dentalcare practice* family (P2) is comprised of three variables: dentalcare frequency (0 = never; 1 = sometimes; 2 = always once; 3 = always twice or more); frequency using brush/branch (0–3); combination of used materials, given the following scores (0 = not applicable/finger; 1 = paste, powder, coal, ash; 2 = branch only; 3 = brush only; 4 = branch + paste/powder/coal/ash; 5 = brush & branch + paste/powder/coal/ ash; 6 = brush + paste/powder/coal/ash; 7 = brush + paste only). *Other hygiene practice* family (P3) comprises P1, P2, and additional *shoe-wearing habits at school* (0–1) and *shoe-wearing habits at home* (inside latrine and in courtyard) (0–4). The *hand cleanliness* family (P4) assessed by the surveyors is composed of *clean hands* (0–1), *trimmed nails* (0 = no; 1 = some; 2 = all), and *clean nails* (0–2). *Hand cleanliness + adenosine*

*triphosphate (ATP)* family (P4E) adds ATP improvement rate comparing before and after handwashing to P4. The ATP luminometer detects actively growing microorganisms measuring the light emitted from the reaction of ATP with the naturally occurring firefly enzyme luciferase. Although smaller figures suggest cleaner hands, i.e., less microorganisms, given large individual variations in the figures, tests were conducted before and after handwashing to measure whether appropriate handwashing had been exercised. The ATP improvement rate was calculated as (ATP-before – ATP-after)/ATP-before; thus, the larger the figures the higher the improvement. The ATP measurements were taken for 10% of child samples and only the endline figures are used for the analysis due to invalid measurement at baseline.

*Nutrition practice* family (P5) contains *breakfast eating habit* (*0 = no; 1 = sometimes; 2 = always*), *food eaten in three days* (*1 = no protein; 2 = protein*), and *breakfast taken*, ordered by the richness of the nutrition score and categorised into seven categories (*1 = none/liquid; 2 = carbohydrate (and fat); 3 = carbohydrate and vitamins; 4 = vegetable/animal protein and vitamins; 5 = vegetable protein or animal protein and carbohydrate; 6 = vegetable protein, animal protein, and carbohydrate; 7 = vegetable protein, animal protein, carbohydrate, and vitamins*). Regarding *health/hygiene knowledge* family (P6), we have *handwashing procedure* explained above, *breakfast important* (*0 = no/don't know; 1 = yes*). For *health/hygiene knowledge + extra* family (P6E), we additionally include outcomes measured only for the endline, *putting water in latrine before defecating*, *oral rehydration solution (ORS) making*, and *food pyramid*, all taking values (*0–2*).

The secondary, more indirect outcome families concern child health improvements as follows: (I1) *cold-related symptoms* at present as well as in the past two-weeks, composed of *cough*, *breathing difficulty*, *sore throat*, *fever*, *running nose*, and *congested nose*; (I2) *other illness*: *diarrhoea*, *stomachache*, and *scabies*, all measured in the past two-weeks, and *fatigue*, *dizziness*, and *appetite loss* all assessed presently; (I3) *anthropometry* which includes *height for age z-score*, *weight for age-z-score* (net of clothes) and *BMI for age z-score*. Outer symptoms such as *cough*, *running/congested nose*, *fever*, *breathing difficulty*, *sore throat* and *scabies* were verified by the enumerators at the time of interview. Indicators that required two-week recall might entail recall bias, although no difference was expected across treatment groups. For all health outcomes, it was measured in (*0–1*).

Regarding the *anthropometric measures* (*z-scores*), given the fact that WHO Growth Chart for weight was only available for 5–10 years, while that for height and BMI is available for 5–19 years, the British 1990 Growth Charts available for 0–20 years were used to calculate the z-scores. The standard flag criteria for the British formula were set at ±5 SD, however, given the general thinness of Bangladeshi children and the majority of outliers having weights between –6 SD and –5 SD, our study set the criteria as –6 SD/ + 5 SD. Both measurements of weight and height were conducted twice by a pair of enumerators to avoid mismeasurements. Technical checks for weight scales were conducted to ensure accuracy and minimise technical errors.

## Exogenous covariates

As for additional controls, we include child *sex*, *age*, *child wealth index* and *parents' literacy*. While child age is also used for anthropometric z-score calculations, recording accurate child age is not a straightforward task, since such records are frequently unavailable in less developed countries. We thus gathered five pieces of age information, in the following order of priority: (1) birth certificate; (2) vaccination card; (3) mother's memory; (4) school register; (5) child's memory. In case of no official child age record and sizable discrepancies among the above (3)~(5) figures, the actual anthropometric measurements were taken into accounts in determining the most probable age information. There were 79 cases without any child age information, all at baseline. Since we followed up the same children in the endline, the baseline age of 68 children could be calculated from the endline data.

*Child wealth index* is created through iterated principal factor, reflecting house structure materials, roof materials, number of rooms, latrine structure and materials, and possession of electronic appliances, mobile phones and bikes. *Parents' literacy* reflects whether the child's mother/father can read/write (*0–4*). Child household wealth and parents' literacy levels are considered, since family wealth and education level may affect child healthy behaviour acquisition, despite

school-based intervention directly targeting the children likely mitigate the impact of family socioeconomic status. The above-cited and another study [41] showed that households in the squatter settlement with lower socioeconomic status took longer—several months—to acquire a healthy habit. On the other hand, another study [42] found that specific information about arsenic in wells prompted costly behaviour of well changes, influenced positively by educational level but not by household wealth.

## Empirical model

Because of its randomised design, we may assume that random assignment of schools to treatment and comparison groups ensures that the schools in either group are similar in all other respects except for having/not receiving a treatment. Nonetheless, there is always a possibility of randomisation being imperfect. Thus, a difference-in-differences (DID) estimation is conducted to adjust for any pre-existing differences between treatment and comparison schools. With $Y_{ijt}$ as an outcome for child $i$ in school $j$ at period $t$, $t$ as the time-period, $T_j$ as the treatment status for school $j$, $\mathbf{X}_{jt}$ as a vector of school-level control variables for $j^{th}$ school, $\mathbf{Z}_{ijt}$ as a vector of child-level control variables, and $D_j$ as a *school-type* dummy which is used for sample stratification of school types, GPS/RNGPS, we estimate the following equation:

$$Y_{ijt} = \alpha + \beta \cdot t \cdot T_j + \delta \cdot t + \gamma \cdot T_j + \varphi \cdot \mathbf{X}_{jt} + \theta \cdot \mathbf{Z}_{ijt} + \eta \cdot D_j + v_{ijt} \tag{1}$$

This makes the estimates of treatment effects $\beta$ more precise and accounts for chance differences between groups in the distribution of pre-randomisation traits. A composite error is $v_{ijt} = c_j + \omega_{ij} + u_{ijt}$, where $c_j$ is unobserved group heterogeneity, $\omega_{ij}$ is unobserved child heterogeneity, and $u_{ijt}$ is the idiosyncratic error.

In addition to the DID formulation, we provide the ITT effects, which reflect the differences between the treatment and control groups. Applying SUR, these effects are presented in terms of family-wise mean-standardised average effect sizes, controlling for all covariates and baseline outcome values (see Supporting Information S1 Table in S1 Zip).

## Handling of intracluster correlation

Given that the RCT was executed at the school level, child-level estimations need to take into account of possible intracluster correlation within school, defined as the ratio of between-cluster variance divided by total variance, which is the sum of within-cluster and between-cluster variance. Hence, the estimations of family-wise mean-standardised average treatment effects apply cluster-robust standard errors (CRSE). However, while the DID estimation with CRSE is the most commonly and widely used method for this kind of multilevel models [43–45], CRSE is not a panacea [38,45]; especially with a single baseline and follow-up, it would require twice the sample size in DID to get the same power as obtained with the analysis of covariance (ANCOVA) [46]. Moreover, in case the treatment effect extending beyond the individual level and affecting the entire cluster, adjusting for such correlation can result in underestimating the true treatment effect. Therefore, in addition to the DID estimation with CRSE within the SUR framework for family-wise outcomes, we conduct a DID estimation without CRSE.

For single outcome estimations, in addition to the DID with CRSE, we also apply an ANCOVA estimation and its likewise alternative, a constrained baseline analysis (CBA), which are both reported to be more efficient than DID-CRSE or post estimator which uses only the endline data [47]. Detailed description of the model, estimated treatment effects and the interpretation for child-level single outcomes are given in the Supporting Information (S4 Table in S1 Zip).

## Results

### General baseline observations

Descriptive statistics for the main variables at baseline for HE-treatment and HE-control groups as well as cross-cutting HESP-treatment groups are provided below. Outcome variables are categorised into families as explained above. Below

we provide general baseline observations of outcome variables as well as of child characteristics, namely, *sex*, *age*, *child wealth index* and *parents' literacy*. Note that while some of the outcomes are measured in a binary term, they are treated numerically in the estimations of average causal treatment effects, reflecting differences in occurrence probabilities, as discussed in Statistical analysis subsection. Due to binary nature of these outcomes, mean values presented in Table 2 directly indicate the proportion of positive responses which are readily interpretable as frequency measures. Four outcome variables which are available only at endline are indicated in italics and the endline descriptive statistics are given for reference.

While randomisation assured statistical indifference across school-level variables, it was not possible to do so across all child-level variables. Nonetheless, only few variables exhibited statistical difference at the 5% significance level, indicated with an asterisk in the p-value column in Table 2. Statistical differences were assessed by the school cluster-adjusted Wald test among the treatment groups. For the HE-treatment groups, only *handwashing with soap before eat* in the primary outcome family *handwashing practice* (P1) was statistically different, with the non-HE (control) group exhibiting higher mean value. For the HESP-treatment groups, *breakfast eaten today* in the *nutrition practice* family (P4) was the sole variable with statistical difference, with the HE-only-treatment group having the highest mean value, and the SP-only-treatment group the lowest. Such non-random baseline differences were dealt by applying the DID estimator.

As for the health-related outcomes, 35% pupils reported having at least one symptom of *cold-related symptoms* (I1) presently. Among cold-related symptoms, cough and running nose were most common, seen in about 20~30% plus children. These symptoms in the past two-weeks were reported by about 40~50% of the children, and fever by about 25%. In terms of other symptoms in the past two-weeks, *diarrhoea*, *stomachache*, *scabies*, *fatigue*, *dizziness* and *appetite loss* (I2 & 3) were experienced by 15.5%, 36.4%, 13.6%, 31.7%, 35.1% and 26.8% of children, respectively. Girls showed higher prevalence of *fatigue*, *dizziness* and *appetite loss* than boys with statistical significance at 0.1% or 1%. These observations seemed to suggest relatively unhealthy children on average. Prevalences of *cold-related symptoms* were mostly higher among the HE-school children, although not statistically different. The average temperature in Dhaka was 24 and 29 Celsius at baseline and endline, respectively. Temperature-wise, cold symptoms were expected to be more prevalent in the baseline survey, although a period dummy would take care of such period-wise differences. While the intracluster correlation $\rho$ was assumed to be 0.058 at the time of sampling calculation, $\rho$s for some outcome variables, such as infectious disease symptoms, were higher than the assumed figure.

In terms of *anthropometric measurements* (I4), all z-scores showed negative values, indicating Bangladeshi children's below international-mean values on average by a standard deviation (SD) of 1.58, 0.84 and 1.56, respectively. For the flagged outlier z-scores, set at –6 SD/+5 SD for weight, height and BMI for this study, there were 5 cases of weight, 2 cases of height, and 14 cases of BMI z-score outliers, which were excluded from the analysis.

## Analysis results and interpretation: SBHE effects

The main results of average-effects model are presented in Table 3 for all children (*all*) estimated with CRSE and without CRSE (*all (no CRSE)*), and the subset of children who were in both baseline and endline surveys (*in both*) estimated with CRSE. The estimated average-effect size (AES) for each of family-wise outcome for the HE-intervention given below mainly refer to all children estimates with CRSE. Statistically significant positive HE-treatment effects were observed for primary outcomes related to hygiene practice and behaviour, namely, (P1) *handwashing* by 0.21 SD [0.12–0.30] (p<0.001), (P2) *dentalcare* by 0.17 SD [0.08–0.27], (p<0.001), (P3) *overall hygiene* by 0.22 SD [0.14–0.31] (p<0.001), (P6) *health/hygiene knowledge* 0.44 SD [0.33–0.55] (p<0.001), and (P6E) *knowledge+extra* 0.20 SD [0.15–0.25] (p<0.001). These significant effects were seen across all three models with similar magnitudes. While (P4) *clean hands*, checked by the surveyor did not find any difference between treatment groups, the non-CRSE estimate for (P4E) *cleans hands+ATP* indicated possible improved cleanliness of hands by handwashing among HE pupils, although significant only

**Table 2. Summary Statistics of Baseline Child Characteristics by the HE-treatment and the Cross-cutting HESP-treatment.**

| | HE-treatment | | | Cross-cutting HESP-treatment | | | | |
|---|---|---|---|---|---|---|---|---|
| | non-HE school | HE school | | control school | SP-only school | HE-only school | HESP-school | |
| | Mean (SD) | Mean (SD) | p-value[a] | Mean (SD) | Mean (SD) | Mean (SD) | Mean (SD) | p-value[a] |
| **Baseline Child Characteristics** | | | | | | | | |
| number of surveyed children | 3,594 | 3,598 | | 1,797 | 1,797 | 1,799 | 1,799 | |
| sex (male:1; female:2) | 1.52 (0.50) | 1.53 (0.50) | 0.174 | 1.51 (0.50) | 1.53 (0.50) | 1.53 (0.50) | 1.54 (0.50) | 0.163 |
| age (year) | 9.15 (1.59) | 9.19 (1.59) | 0.642 | 9.07 (1.57) | 9.24 (1.60) | 9.20 (1.60) | 9.17 (1.59) | 0.322 |
| child (family) wealth index[b] | -0.09 (0.87) | -0.13 (0.83) | 0.378 | -0.06 (0.91) | -0.13 (0.84) | -0.14 (0.82) | -0.13 (0.83) | 0.730 |
| parents' literacy (read) | 1.03 (0.85) | 1.02 (0.86) | 0.826 | 1.05 (0.84) | 1.02 (0.86) | 1.05 (0.86) | 1.00 (0.86) | 0.718 |
| parents' literacy (write) | 1.02 (0.85) | 1.01 (0.86) | 0.717 | 1.04 (0.84) | 1.01 (0.86) | 1.04 (0.86) | 0.98 (0.86) | 0.583 |
| **Primary Outcomes by Families** | | | | | | | | |
| **P1: handwashing practice** | | | | | | | | |
| handwashing with soap before eating | 0.19 (0.39) | 0.14 (0.35) | 0.042* | 0.20 (0.40) | 0.19 (0.39) | 0.14 (0.35) | 0.15 (0.36) | 0.217 |
| handwashing with soap after defecation | 0.65 (0.48) | 0.62 (0.48) | 0.328 | 0.65 (0.48) | 0.65 (0.48) | 0.63 (0.48) | 0.62 (0.48) | 0.810 |
| handwashing with soap after playing | 0.09 (0.29) | 0.07 (0.25) | 0.138 | 0.10 (0.30) | 0.08 (0.27) | 0.06 (0.23) | 0.08 (0.27) | 0.138 |
| handwashing index before eating[c] | 3.17 (1.49) | 3.06 (1.36) | 0.361 | 3.15 (1.52) | 3.19 (1.45) | 3.10 (1.31) | 3.03 (1.40) | 0.785 |
| handwashing index after defecation[c] | 4.79 (1.31) | 4.72 (1.30) | 0.302 | 4.74 (1.33) | 4.84 (1.29) | 4.73 (1.28) | 4.71 (1.33) | 0.529 |
| handwashing index after playing[c] | 2.25 (1.46) | 2.27 (1.37) | 0.768 | 2.20 (1.47) | 2.29 (1.44) | 2.38 (1.32) | 2.17 (1.40) | 0.394 |
| handwashing under running water | 0.99 (0.22) | 0.97 (0.22) | 0.155 | 0.98 (0.22) | 0.99 (0.21) | 0.97 (0.23) | 0.98 (0.21) | 0.474 |
| handwashing procedures[d] | 1.41 (0.80) | 1.38 (0.85) | 0.642 | 1.43 (0.79) | 1.40 (0.80) | 1.42 (0.85) | 1.34 (0.84) | 0.834 |
| **P2: dentalcare practice** | | | | | | | | |
| dentalcare frequency[e] | 10.08 (5.68) | 9.88 (5.33) | 0.395 | 10.36 (5.70) | 9.80 (5.66) | 9.69 (5.39) | 10.06 (5.27) | 0.240 |
| toothbrush/branch use frequency[e] | 1.49 (0.97) | 1.47 (0.93) | 0.490 | 1.54 (0.95) | 1.45 (0.97) | 1.43 (0.94) | 1.50 (0.92) | 0.216 |
| dentalcare materials[f] | 4.53 (2.25) | 4.56 (2.24) | 0.801 | 4.63 (2.20) | 4.44 (2.29) | 4.46 (2.28) | 4.65 (2.20) | 0.196 |
| **P3: overall hygiene practice (+ P1~P3)** | | | | | | | | |
| wearing shoes at school | 0.88 (0.33) | 0.85 (0.35) | 0.191 | 0.87 (0.34) | 0.89 (0.32) | 0.85 (0.36) | 0.86 (0.34) | 0.406 |
| wearing shoes at home (latrine + yard) | 2.97 (1.11) | 2.92 (1.14) | 0.352 | 2.97 (1.13) | 2.98 (1.09) | 2.89 (1.16) | 2.96 (1.13) | 0.622 |
| **P4: hand cleanliness** | | | | | | | | |
| clean hands | 0.48 (0.50) | 0.51 (0.50) | 0.169 | 0.46 (0.50) | 0.50 (0.50) | 0.51 (0.50) | 0.50 (0.50) | 0.277 |
| trimmed nails | 0.79 (0.92) | 0.81 (0.93) | 0.680 | 0.82 (0.93) | 0.76 (0.91) | 0.84 (0.94) | 0.78 (0.92) | 0.640 |
| clean nails | 0.57 (0.85) | 0.58 (0.85) | 0.640 | 0.59 (0.86) | 0.54 (0.83) | 0.59 (0.86) | 0.58 (0.85) | 0.779 |
| *P4E: hand cleanliness ATP (endline only)* | | | | | | | | |
| *number of surveyed children* | *430* | *437* | | *216* | *214* | *216* | *221* | |
| *ATP improvements (endline)* | *0.07 (3.64)* | *0.26 (1.17)* | *0.301* | *0.29 (0.53)* | *-0.15 (5.13)* | *0.35 (0.67)* | *0.18 (1.51)* | *0.220* |
| **P5: nutrition practice** | | | | | | | | |
| breakfast habit | 1.89 (0.34) | 1.91 (0.30) | 0.106 | 1.89 (0.34) | 1.89 (0.35) | 1.92 (0.29) | 1.90 (0.31) | 0.237 |
| breakfast eaten today[g] | 3.82 (1.16) | 3.86 (1.12) | 0.217 | 3.85 (1.18) | 3.78 (1.13) | 3.91 (1.12) | 3.81 (1.12) | 0.044* |
| food taken in 3 days[h] | 1.93 (0.26) | 1.94 (0.23) | 0.060 | 1.93 (0.25) | 1.92 (0.27) | 1.94 (0.24) | 1.95 (0.22) | 0.242 |
| **P6: health knowledge** | | | | | | | | |
| breakfast important | 0.59 (0.49) | 0.54 (0.50) | 0.069 | 0.57 (0.49) | 0.60 (0.49) | 0.54 (0.50) | 0.54 (0.50) | 0.285 |
| handwashing procedures | 1.41 (0.80) | 1.38 (0.85) | 0.642 | 1.43 (0.79) | 1.40 (0.80) | 1.42 (0.85) | 1.34 (0.84) | 0.834 |
| *P6E: health knowledge (endline only)* | | | | | | | | |
| *number of surveyed children* | *4,494* | *4,497* | | *2,247* | *2,247* | *2,250* | *2,247* | |
| *water latrine before (endline)* | *0.03 (0.18)* | *0.03 (0.17)* | *0.922* | *0.02 (0.16)* | *0.03 (0.19)* | *0.02 (0.17)* | *0.03 (0.17)* | *0.927* |

*(Continued)*

**Table 2.** (Continued)

| | HE-treatment | | | Cross-cutting HESP-treatment | | | | |
| --- | --- | --- | --- | --- | --- | --- | --- | --- |
| | non-HE school | HE school | | control school | SP-only school | HE-only school | HESP-school | |
| | Mean (SD) | Mean (SD) | p-value[a] | Mean (SD) | Mean (SD) | Mean (SD) | Mean (SD) | p-value[a] |
| *food pyramid (endline)* | *0.58 (0.85)* | *0.82 (0.92)* | *<0.001* | *0.62 (0.88)* | *0.54 (0.83)* | *0.83 (0.93)* | *0.80 (0.92)* | *<0.001* |
| *ORS making (endline)* | *1.03 (0.82)* | *1.10 (0.83)* | *0.088* | *1.09 (0.82)* | *0.97 (0.82)* | *1.10 (0.83)* | *1.10 (0.83)* | *0.021* |
| **Secondary Outcomes by Families** | | | | | | | | |
| **I1: cold-related symptoms** | | | | | | | | |
| cough | 0.22 (0.42) | 0.23 (0.42) | 0.730 | 0.23 (0.42) | 0.22 (0.41) | 0.22 (0.42) | 0.23 (0.42) | 0.919 |
| breathing difficulty | 0.01 (0.10) | 0.01 (0.11) | 0.561 | 0.01 (0.10) | 0.01 (0.11) | 0.01 (0.10) | 0.02 (0.13) | 0.735 |
| sore throat | 0.01 (0.07) | 0.01 (0.08) | 0.393 | 0.01 (0.09) | 0.00 (0.05) | 0.01 (0.07) | 0.01 (0.10) | 0.069 |
| fever | 0.07 (0.25) | 0.08 (0.27) | 0.160 | 0.07 (0.25) | 0.07 (0.26) | 0.08 (0.28) | 0.08 (0.27) | 0.521 |
| running nose | 0.31 (0.46) | 0.32 (0.47) | 0.440 | 0.30 (0.46) | 0.31 (0.46) | 0.32 (0.47) | 0.32 (0.47) | 0.879 |
| congested nose | 0.05 (0.22) | 0.04 (0.21) | 0.487 | 0.05 (0.22) | 0.05 (0.21) | 0.04 (0.20) | 0.05 (0.21) | 0.877 |
| cough within 2 weeks | 0.38 (0.49) | 0.41 (0.49) | 0.249 | 0.39 (0.49) | 0.38 (0.49) | 0.41 (0.49) | 0.41 (0.49) | 0.711 |
| breathing difficulty within 2 weeks | 0.02 (0.14) | 0.03 (0.16) | 0.378 | 0.02 (0.15) | 0.02 (0.12) | 0.02 (0.14) | 0.03 (0.17) | 0.404 |
| sore throat within 2 weeks | 0.02 (0.13) | 0.02 (0.15) | 0.316 | 0.02 (0.14) | 0.02 (0.12) | 0.02 (0.13) | 0.03 (0.17) | 0.310 |
| fever within 2 weeks | 0.25 (0.43) | 0.26 (0.44) | 0.575 | 0.23 (0.42) | 0.27 (0.44) | 0.27 (0.44) | 0.25 (0.43) | 0.262 |
| running nose within 2 weeks | 0.48 (0.50) | 0.51 (0.50) | 0.109 | 0.47 (0.50) | 0.48 (0.50) | 0.52 (0.50) | 0.51 (0.50) | 0.400 |
| congested nose within 2 weeks | 0.09 (0.29) | 0.10 (0.29) | 0.729 | 0.10 (0.30) | 0.08 (0.27) | 0.09 (0.28) | 0.10 (0.31) | 0.568 |
| **I2: infectious disease (+I2)** | | | | | | | | |
| diarrhoea within 2 weeks | 0.15 (0.36) | 0.16 (0.36) | 0.587 | 0.15 (0.36) | 0.15 (0.36) | 0.17 (0.37) | 0.15 (0.36) | 0.645 |
| stomachache within 2 weeks | 0.36 (0.48) | 0.37 (0.48) | 0.876 | 0.40 (0.49) | 0.33 (0.47) | 0.37 (0.48) | 0.36 (0.48) | 0.154 |
| scabies | 0.14 (0.35) | 0.13 (0.34) | 0.367 | 0.15 (0.36) | 0.14 (0.34) | 0.12 (0.33) | 0.14 (0.34) | 0.624 |
| **I3: overall health (+I1, I2)** | | | | | | | | |
| fatigue | 0.33 (0.47) | 0.31 (0.46) | 0.385 | 0.34 (0.47) | 0.31 (0.46) | 0.31 (0.46) | 0.31 (0.46) | 0.646 |
| dizziness | 0.36 (0.48) | 0.35 (0.48) | 0.602 | 0.36 (0.48) | 0.35 (0.48) | 0.35 (0.48) | 0.34 (0.47) | 0.819 |
| appetite loss | 0.27 (0.44) | 0.27 (0.44) | 0.845 | 0.27 (0.44) | 0.27 (0.45) | 0.26 (0.44) | 0.27 (0.45) | 0.974 |
| **I4: anthropometry[i]** | | | | | | | | |
| height z-score[i] | -1.58 (1.23) | -1.57 (1.20) | 0.778 | -1.51 (1.23) | -1.66 (1.22) | -1.57 (1.23) | -1.57 (1.17) | 0.251 |
| weight z-score (net of clothes)[i] | -0.84 (1.24) | -0.83 (1.23) | 0.813 | -0.76 (1.26) | -0.92 (1.22) | -0.86 (1.22) | -0.79 (1.24) | 0.250 |
| BMI z-score[i] | -1.56 (1.06) | -1.55 (1.02) | 0.729 | -1.53 (1.08) | -1.59 (1.04) | -1.52 (1.05) | -1.58 (0.98) | 0.121 |

Notes: Mean values are presented for binary and ordinal outcomes since they are treated as numerical variables in the estimation of average causal treatment effect which exhibits differences in probabilities, as discussed in Statistical analysis subsection above. (a) school cluster-adjusted Wald test with significant differences between groups noted with asterisk (* p < 0.05, ** p < 0.01, *** p < 0.001); (b) child family wealth index is created through iterated principal factor, reflecting house structure materials, roof materials, number of rooms, latrine structure and materials, possession of electronic appliances, phones and bikes; (c) index reflects handwashing frequencies on each occasion and usage of soap, ash, mud, only water on each occasion (0–8); (d) correct handwashing procedure (0–8); (e) frequency of dentalcare (never, sometimes, always; once a day, twice a day) (0–3); (f) dentalcare material combinations (fingers, branch to brush, with ash, coal, powder and/or paste) (0–7); (g) ordered by the richness of the nutrition score (1–7) (none; carbohydrate (and fat); carbohydrate and vitamins; vegetable/animal protein and vitamins; vegetable/animal protein and carbohydrate; vegetable protein, animal protein and carbohydrate; protein, carbohydrate and vitamins); (h) no protein, protein (1–2); (i) the British 1990 Growth Charts are used to calculate the anthropometric z-scores; 5, 2, 14 outliers omitted for weight, height, BMI z-scores, respectively.

at the 10% level. Given the high variability in individual ATP measurements and the small sample size, these findings still provide valuable initial insights.

For secondary outcomes, HE-treatment had a beneficial effect only on (I1) *cold-related symptoms*, denoted by a negative coefficient of −0.05. When estimated with CRSE, the confidence interval was [−0.10–0.01] (p = 0.086), indicating

**Table 3. Family-wise mean-standardised HE-treatment effect in average effect size on nine outcome families for all children and children in both surveys.**

**Primary Outcomes**

| | (P1) handwashing | | | (P2) dentalcare | | | (P3) overall hygiene | | |
|---|---|---|---|---|---|---|---|---|---|
| | AES-coefficient [95%CI] [p-value] | | | AES-coefficient [95%CI] [p-value] | | | AES-coefficient [95%CI] [p-value] | | |
| | all | all (no CRSE) | in both | all | all (no CRSE) | in both | all | all (no CRSE) | in both |
| HE | 0.21*** | 0.22*** | 0.23*** | 0.17*** | 0.17*** | 0.17*** | 0.22*** | 0.22*** | 0.24*** |
| | [0.12, 0.30] | [0.18, 0.25] | [0.14, 0.31] | [0.08, 0.27] | [0.12, 0.22] | [0.07,.0.27] | [0.14, 0.31] | [0.19, 0.25] | [0.15, 0.32] |
| | [0.000] | [0.000] | [0.000] | [0.000] | [0.000] | [0.001] | [0.000] | [0.000] | [0.000] |
| HE-group | −0.05 | −0.05*** | −0.06 | −0.03 | −0.03 | −0.03 | −0.06+ | −0.06*** | −0.07* |
| | [-0.13, 0.03] | [-0.07, -0.03] | [-0.13, 0.01] | [-0.10, 0.05] | [-0.07, 0.01] | [-0.11, 0.04] | [-0.12, 0.00] | [-0.08, -0.04] | [-0.13, -0.01] |
| | [0.200] | [0.000] | [0.112] | [0.486] | [0.190] | [0.413] | [0.069] | [0.000] | [0.033] |
| period | 0.64*** | 0.64*** | 0.67*** | 0.36*** | 0.36*** | 0.39*** | 0.53*** | 0.53*** | 0.57*** |
| | [0.58, 0.71] | [0.61, 0.66] | [0.61, 0.73] | [0.29, 0.42] | [0.32, 0.39] | [0.32, 0.46] | [0.47, 0.59] | [0.51, 0.55] | [0.51, 0.63] |
| | [0.000] | [0.000] | [0.000] | [0.000] | [0.000] | [0.000] | [0.000] | [0.000] | [0.000] |
| school type | −0.08** | −0.07*** | −0.08** | −0.10** | −0.10*** | −0.10*** | −0.08** | −0.08*** | −0.09*** |
| | [-0.13, -0.02] | [-0.09, -0.05] | [-0.13, -0.02] | [-0.16, -0.04] | [-0.12, -0.08] | [-0.16, -0.04] | [-0.13, -0.03] | [-0.10, -0.07] | [-0.14, -0.04] |
| | [0.008] | [0.000] | [0.004] | [0.002] | [0.000] | [0.001] | [0.001] | [0.000] | [0.000] |
| sex | 0.10*** | 0.09*** | 0.09*** | 0.04** | 0.04*** | 0.04** | 0.08*** | 0.08*** | 0.07*** |
| | [0.08, 0.12] | [0.07, 0.11] | [0.07, 0.11] | [0.01, 0.07] | [0.02, 0.07] | [0.01, 0.07] | [0.06, 0.09] | [0.06, 0.09] | [0.06, 0.09] |
| | [0.000] | [0.000] | [0.000] | [0.002] | [0.001] | [0.008] | [0.000] | [0.000] | [0.000] |
| N | 16181 | 16181 | 12224 | 16183 | 16183 | 12226 | 16163 | 16163 | 12211 |

| | (P4) clean hands | | | (P4E) clean hands+ATP (endline) | | | (P5) nutrition | | |
|---|---|---|---|---|---|---|---|---|---|
| | AES-coefficient [95%CI] [p-value] | | | AES-coefficient [95%CI] [p-value] | | | AES-coefficient [95%CI] [p-value] | | |
| | all | all (no CRSE) | in both | all | all (no CRSE) | in both | all | all (no CRSE) | in both |
| HE | −0.01 | −0.01 | −0.01 | 0.08 | 0.08+ | 0.08 | −0.04 | −0.04* | −0.05+ |
| | [-0.11, 0.10] | [-0.06, 0.04] | [-0.12, 0.10] | [-0.02, 0.17] | [-0.01, 0.16] | [-0.03, 0.19] | [-0.10, 0.01] | [-0.08, -0.00] | [-0.10, 0.01] |
| | [0.906] | [0.802] | [0.871] | [0.122] | [0.076] | [0.175] | [0.122] | [0.030] | [0.080] |
| HE-group | 0.03 | 0.03+ | 0.04 | | | | 0.06* | 0.06*** | 0.06** |
| | [-0.04, 0.11] | [-0.00, 0.07] | [-0.04, 0.11] | | | | [0.01, 0.10] | [0.03, 0.08] | [0.02, 0.11] |
| | [0.412] | [0.086] | [0.368] | | | | [0.012] | [0.000] | [0.007] |
| period | 0.40*** | 0.40*** | 0.45*** | | | | −0.01 | −0.01 | 0.007 |
| | [0.31, 0.48] | [0.36, 0.43] | [0.36, 0.53] | | | | [-0.05, 0.03] | [-0.04, 0.02] | [-0.03, 0.05] |
| | [0.000] | [0.000] | [0.000] | | | | [0.604] | [0.475] | [0.725] |
| school type | −0.04 | −0.04** | −0.04 | 0.01 | 0.01 | 0.03 | −0.01 | −0.01 | −0.01 |
| | [-0.10, 0.02] | [-0.07, -0.02] | [-0.10, 0.02] | [-0.09, 0.10] | [-0.08, 0.09] | [-0.08, 0.14] | [-0.04, 0.02] | [-0.03, 0.01] | [-0.04, 0.02] |
| | [0.184] | [0.002] | [0.173] | [0.870] | [0.851] | [0.620] | [0.423] | [0.252] | [0.387] |
| sex | 0.15*** | 0.15*** | 0.16*** | 0.06 | 0.06 | 0.04 | 0.04*** | 0.04*** | 0.04** |
| | [0.12, 0.18] | [0.12, 0.17] | [0.13, 0.20] | [-0.02, 0.15] | [-0.02, 0.15] | [-0.05, 0.14] | [0.02, 0.06] | [0.02, 0.06] | [0.01, 0.06] |
| | [0.000] | [0.000] | [0.000] | [0.136] | [0.144] | [0.391] | [0.000] | [0.000] | [0.002] |
| N | 16178 | 16178 | 12221 | 867 | 867 | 600 | 16183 | 16183 | 12226 |

| | (P6) knowledge | | | (P6E) knowledge+extra (endline) | | | | | |
|---|---|---|---|---|---|---|---|---|---|
| | AES-coefficient [95%CI] [p-value] | | | AES-coefficient [95%CI] [p-value] | | | | | |
| | all | all (no CRSE) | in both | all | all (no CRSE) | in both | | | |
| HE | 0.44*** | 0.44*** | 0.45*** | 0.20*** | 0.20*** | 0.21*** | | | |

*(Continued)*

**Table 3.** (Continued)

**Primary Outcomes**

| | (P1) handwashing AES-coefficient [95%CI] [p-value] | | | (P2) dentalcare AES-coefficient [95%CI] [p-value] | | | (P3) overall hygiene AES-coefficient [95%CI] [p-value] | | |
|---|---|---|---|---|---|---|---|---|---|
| | all | all (no CRSE) | in both | all | all (no CRSE) | in both | all | all (no CRSE) | in both |
| | [0.33, 0.55] | [0.39, 0.48] | [0.34, 0.56] | [0.15, 0.25] | [0.18, 0.22] | [0.15, 0.26] | | | |
| | [0.000] | [0.000] | [0.000] | [0.000] | [0.000] | [0.000] | | | |
| HE-group | −0.08 | −0.08*** | −0.06 | | | | | | |
| | [-0.18, 0.02] | [-0.11, -0.04] | [-0.16, 0.03] | | | | | | |
| | [0.130] | [0.000] | [0.204] | | | | | | |
| period | 0.80*** | 0.80*** | 0.84*** | | | | | | |
| | [0.72, 0.88] | [0.77, 0.83] | [0.76, 0.92] | | | | | | |
| | [0.000] | [0.000] | [0.000] | | | | | | |
| school type | −0.02 | −0.03* | −0.03 | −0.04 | −0.04** | −0.03 | | | |
| | [-0.09, 0.04] | [-0.05, -0.00] | [-0.10, 0.04] | [-0.09, 0.02] | [-0.06, -0.01] | [-0.09, 0.02] | | | |
| | [0.433] | [0.022] | [0.375] | [0.171] | [0.002] | [0.244] | | | |
| sex | 0.03* | 0.03* | 0.02 | 0.03* | 0.03** | 0.02 | | | |
| | [0.00, 0.05] | [0.00, 0.05] | [-0.00, 0.05] | [0.01, 0.05] | [0.01, 0.05] | [-0.00, 0.05] | | | |
| | [0.023] | [0.017] | [0.107] | [0.016] | [0.009] | [0.112] | | | |
| N | 16183 | 16183 | 12226 | 8991 | 8991 | 6113 | | | |

**Secondary Outcomes**

| | (I1) cold-related symptoms AES-coefficient [95%CI] [p-value] | | | (I2) other illnesses AES-coefficient [95%CI] [p-value] | | | (I3) anthropometry AES-coefficient [95%CI] [p-value] | | |
|---|---|---|---|---|---|---|---|---|---|
| | all | all (no CRSE) | in both | all | all (no CRSE) | in both | all | all (no CRSE) | in both |
| HE | −0.05+ | −0.05** | −0.05+ | 0.02 | 0.02 | −0.004 | −0.02 | −0.02 | −0.02 |
| | [-0.10, 0.01] | [-0.08, -0.01] | [-0.10, 0.01] | [-0.04, 0.08] | [-0.02, 0.05] | [-0.07, 0.06] | [-0.06, 0.02] | [-0.08, 0.03] | [-0.05, 0.01] |
| | [0.086] | [0.007] | [0.078] | [0.573] | [0.324] | [0.906] | [0.262] | [0.416] | [0.107] |
| HE-group | 0.03 | 0.030* | 0.029 | −0.02 | −0.02 | −0.01 | 0.01 | 0.01 | 0.04 |
| | [-0.01, 0.07] | [0.01, 0.05] | [-0.01, 0.07] | [-0.08, 0.04] | [-0.05, 0.01] | [-0.07, 0.05] | [-0.06, 0.08] | [-0.03, 0.05] | [-0.03, 0.10] |
| | [0.166] | [0.015] | [0.177] | [0.569] | [0.227] | [0.719] | [0.756] | [0.592] | [0.279] |
| period | −0.05* | −0.05*** | −0.05* | −0.19*** | −0.19*** | −0.16*** | 0.14*** | 0.14*** | 0.18*** |
| | [-0.09, -0.00] | [-0.07, -0.02] | [-0.09, -0.00] | [-0.23, -0.14] | [-0.21, -0.16] | [-0.21, -0.11] | [0.11, 0.17] | [0.10, 0.18] | [0.16, 0.20] |
| | [0.032] | [0.000] | [0.042] | [0.000] | [0.000] | [0.000] | [0.000] | [0.000] | [0.000] |
| school type | −0.003 | −0.003 | −0.003 | −0.01 | −0.01 | −0.01 | −0.01 | −0.01 | −0.003 |
| | [-0.03, 0.03] | [-0.02, 0.01] | [-0.04, 0.03] | [-0.05, 0.03] | [-0.02, 0.01] | [-0.05, 0.03] | [-0.07, 0.06] | [-0.03, 0.02] | [-0.07, 0.06] |
| | [0.830] | [0.699] | [0.880] | [0.735] | [0.437] | [0.692] | [0.846] | [0.656] | [0.929] |
| sex | 0.002 | 0.002 | 0.003 | 0.08*** | 0.08*** | 0.07*** | −0.05** | −0.05** | −0.04+ |
| | [-0.02, 0.02] | [-0.02, 0.02] | [-0.02, 0.02] | [0.06, 0.09] | [0.06 0.09] | [0.05, 0.09] | [-0.09, -0.01] | [-0.08, -0.02] | [-0.08, 0.00] |
| | [0.838] | [0.830] | [0.768] | [0.000] | [0.000] | [0.000] | [0.008] | [0.000] | [0.065] |
| N | 16183 | 16183 | 12226 | 16173 | 16173 | 12217 | 16130 | 16130 | 12207 |

Notes: Each column represents a separate regression on a family of outcomes applying seemingly unrelated regressions (SUR), estimated by a feasible generalised least squares (FGLS) estimator with cluster-robust standard errors (CRSE) unless stated otherwise. β-coefficient is the mean-standardised effect from SUR applying the difference-in-differences (DID) model, controlling for school type and child sex. For each outcome family, estimates are provided for all sample children and children present *in both* baseline and endline. Each indicator family includes the following variables: (P1) *handwashing practice*: handwashing washing frequency in each occasion (before eating, after defecation, after playing), used substances (soap, ash, mud and/or water only), washing with soap in each occasion, wash with running water, correct washing procedure; (P2) *dentalcare practice*: frequency of dentalcare, frequency of using brush/branch, type of materials used; (P3) *overall hygiene practice*: shoes/footwear wearing at school (frequency), shoes/footwear

*(Continued)*

**Table 3.** (Continued)

wearing at home (frequency in latrine and in courtyard), +P1 & P2; (P4) *clean hand*s: clean hands by observation, trimmed nails, clean nails; (P4E) *clean hand*s+*ATP*: additional hand cleanliness measured by ATP improvement rate (10% of samples); (P5) *nutrition practice*: breakfast habit, breakfast taken in 3 days, food taken in 3 days, ordered by the richness of nutrition score (none; carbohydrate (and fat); carbohydrate and vitamins; vegetable/animal protein and vitamins; vegetable/animal protein and carbohydrate; vegetable protein, animal protein and carbohydrate; protein, carbohydrate and vitamins); (P6) *health/hygiene knowledge*: handwashing procedure, breakfast significance; (P6E) *health/hygiene knowledge+extra:* additional knowledge measured only in the endline, i.e., putting water in latrine before defecating, oral rehydration solution (ORS) making, food pyramid; (I1) *cold-related symptoms*: symptoms at present and in the past two-weeks, cough, breathing difficulty, sore throat, fever, running nose, congested nose; (I2) *other illness*: diarrhoea, stomachache, skin disease, fatigue, dizziness, appetite loss in the past two-weeks; (I3) *anthropometry*: height-, weight-, BMI-z-score;. Significance level: +p<0.1, * p<0.05, ** p<0.01, ***p<0.001; 95% confidence intervals and p-value in brackets. The results are robust with additional adjustment with *child age*, *wealth index*, and *parent literacy* (see S2 Table in S1 Zip).

marginal significance at the 10% level. Without CRSE, the confidence interval tightened to [−0.08, −0.01] (p=0.007), achieving statistical significance at the 1% level. This discrepancy likely result from the high intracluster correlation given their infectious nature, which CRSE accounts for by adjusting the standard errors. Moreover, the intracluster correlation for all *cold-related symptoms* family was significantly reduced from a baseline of $\rho=0.063$ to an endline of $\rho=0.037$ (p=0.012). The absolute reduction of intracluster correlation was significant only for the HE-treatment group at 0.30 [0.004, 0.06] (p=0.021), in comparison with 0. 22 [−0.01, 0.05] (p=0.171) for the control group. The residual intracluster correlation for *cold related symptoms* after the mixed-effect analysis was 0.29 [0.21, 0.04]. Given that seasonal effects have been accounted for by the inclusion of period control, the observed reduction in both cold symptoms and intracluster correlations suggest a positive shift in school-level dynamics, particularly among the HE-treatment schools. This shift likely reflects a reduced prevalence of cold symptoms due to increased awareness, better hygiene practices, and an overall improvement in the school health environment.

The above findings were robust to the exclusion of *sex*, as well as the inclusion of extra controls, *child age*, *child wealth index*, and *parents' literacy* which themselves had statistically significant positive effects on most of the outcome families (estimation results with additional covariates in the S2 Table in S1 Zip). Estimated positive *sex* coefficients indicate that girls had better hygiene practice, yet negative coefficients for *anthropometry* and positive coefficients for *other illness* indicate that girls had worse health status. Adding *child age* did not affect the HE-effect estimates but slightly reduced the magnitude of period effects. Child age had statistically significant positive effects on all primary outcomes while it had statistically significant negative effects on all secondary outcomes. In terms of illness-related secondary outcomes, this suggests that older pupils were likely healthier than younger pupils.

*Period* was statistically significant at the 0.1% level for all estimation apart from *nutrition practice*. Negative *period* coefficients in *cold-related symptoms* and *other illness* demonstrated improved health from baseline to endline. While period effects on *cold-related symptoms* could encompass seasonality effects as noted above, those on *other illness* were not obviously related to seasons. For other health-KAPB relevant outcomes, *period* exhibited higher positive effects than *HE-treatment*, signalling general improvement in health-KAPB across pupils since the baseline, regardless of treatment status. Statistically significant negative effects of *school type* on *handwashing practice*, *dentalcare practice* and *other hygiene practice* suggested that pupils going to GPS school fared better in these aspects. Negative effects of school type on these outcome families indicate that RNGPS pupils had less of healthy practices. As a matter of fact, not only the government school funding level but also the child household wealth indicator statistically differed between these school types. Thus, the type of school seems to be correlated with pupils' socioeconomic status.

Selected results of cross-cutting HESP-intervention analysis are given in the S3 Table in in S1 Zip for models where HE-treatment had statistically significant effects above. Briefly summarising the estimates, HE-only-treatment had positive effect with statistical significance on these outcomes, namely, *handwashing*, *dentalcare*, *overall hygiene*, and *health/hygiene knowledge*. Similarly, HESP-treatment positively affected these outcomes, except for *dentalcare*. Unlike HE-only-treatment and HESP-treatment, SP-only-treatment had no positive impact. The HESP-treatment had higher impact

than HE-only treatment on *handwashing* and *overall hygiene*, indicating possible additional positive effect of soap availability. Particularly for *handwashing*, *overall hygiene*, and *health/hygiene knowledge*, the sum of HE-only and HESP-treatment effects were higher than that of HE-treatment in Table 3.

Single outcome estimations of HE-treatment effects are given in the S3 Table in S1 Zip.

## Health education spillover effects

As noted above, *period* had significant positive coefficient with large magnitudes regardless of treatment status for all outcomes but *nutrition practice*. We can speculate several reasons for this: (1) there were general improvements in health practices and hygiene status across all schools due to external infrastructural improvements (but not general nutrition level); (2) there were certain HE-externality effects across schools in terms of skill and/or information transmission. While general health improvements are plausible along with economic development, improvements in health KABP are suggestive of inter-school spillovers, possibly including John Henry effects among the control schools in trying to become the next intervention targets.

There were several factors that could have induced inter-school spillover effects. One was school proximities; schools were distributed in two sub-districts and their mean distance among two schools were 15 km, while minimum distance was 0.3 km and maximum was 41.4 km (see school location map in S1 Fig in in S1 Zip). The average and maximum number of schools within 2 km, 3 km and 6 km radius of the HE schools were 1.7 and 6 schools, 3.9 and 10 schools, and 13.7 and 28 schools, respectively. In the area with several schools in proximity, it was not unusual for children from neighbouring households to go to different schools. In such circumstances, skills and knowledge they had learned in HE sessions might have been talked of. On the other hand, despite being in an era of communication technology, health skills which children learnt at schools would not be a likely topic discussed over mobile phones by their guardians, nor were such skills easily transmittable this way. Although the project implementation attempted to mitigate possible John Henry effects among non-treatment schools as described above, such effects and consequent spillovers might have arisen especially given that HE-skills and knowledge were not complicated per se. Even for schools not in proximities, there were occasional school headteachers meetings and other casual teacher meetings where some conversation over this external project could have taken place. Thus, information transmission barrier could be low if sufficient interests existed, although practicing it and having behaviour change would be different. Examples of informational spillovers in field experiments abound [48–51]. We thus investigate whether there were any such spillovers, and if so, how they might have affected the treatment outcomes.

## Empirical model with an externality term

We add a variable to capture possible inter-school spillover effects to the above estimation equations. Such externalities are captured by distances between schools, the number of schools in proximity, and the number of pupils attending the school since HE-school pupils might have spoken about the HE contents to pupils from other schools. In the influential work by [17], inter-school externalities of deworming drug effects depended on the sums of the total number of students at school within a certain distance of a treatment school. In a recent work by [51], the spillover effects of agricultural advisory services were captured by the ratio of farmers who benefit via their social network.

Given that there is no clear theory or model construct as to how such information and skills/practices spread across distances and entities, the following externality variables are considered here: (1) a composite externality index which incorporates distances from the HE-treatment schools as well as the number of attending pupils; (2) the number of schools within certain radius of the HE-treatment schools (as well as in-between-distances). The composite externality index takes the following form: $\frac{1}{J \cdot \mu} \cdot \sum_k lnN_k^T \cdot lnN_j \cdot e^{-d_{kj}}$, where $d_k$ is the distance of school $j$ from a treatment school $k$ for $j \neq k$, whose effect manifests exponential decay—the further the distance, the quicker the effects decay—multiplied by the natural logarithm of the total number of attending students in school $j$, $lnN_j$, and that in the treatment school $k$, $lnN_k^T$, both measured at baseline, summed up for all $K$ treatment schools, and divided by total number of schools $J$ times $\mu$, the natural logarithm

of the average total attending students of all $J$ schools at baseline, in order to normalise. Distances are calculated using global positioning system data of each school location embedded in the photos taken during the survey. For $e^{-d}$, the closer the schools, the lager the figure, and thus the spillover effect, e.g., for 500 m the figure is 0.606, while for 3 km and 6 km, it is 0.050 and 0.002, respectively. We use the baseline students' data because using the endline data would create an endogeneity problem, and we use attending students' data rather than admitted students' data because the latter can be significantly different from actual number of pupils. It was a routine practice especially for RNGPS schools to inflate the number of admitted students in order to obtain more school funds (a pairwise correlation between admitted and attended pupils for GPS and RNGPS at baseline were 0.933 and 0.819, respectively). The number of students applies a log form because an increase in the number of contacts does not increase proportional to the number of pupils, given an increased probability of contacting the same person. Thus, incorporating the treatment externality index, the DID estimation equation (1) becomes:

$$Y_{ijt} = \alpha + \beta \cdot t \cdot T_j + \delta \cdot t + \gamma \cdot T_j + \varphi \cdot \mathbf{X}_{jt} + \theta \cdot \mathbf{Z}_{ijt} + \lambda \cdot \frac{1}{J \cdot \mu} \cdot \sum_k lnN_k^T \cdot lnN_j \cdot e^{-d_{kj}} + \eta \cdot D_j + v_{ijt}. \qquad (2)$$

Here, $\lambda$ is expected to reflect the magnitude of treatment externalities across schools, intensified by the number of attending students and closeness of schools both in a diminishing manner. The extent of externalities, that is, health-related information children learnt at the HE schools spreading beyond their families, thus depends on the number of students and the distance between the schools for the composite externality index.

## Externality/spillover general observations

Concerning the possibility of HE information dispersed by children, 2.6% of HE schools pupils reported to have communicated what they had learnt in SBHE with their school mates, 16.9% of them with their family members, 0.7% with pupils from other schools. Although children may have talked about the HE matters unconsciously and not remembering it, this observation suggests that informational externality among children might not have been large. As noted, online information diffusion was neither likely among children nor parents. While there could be some informational spillover through family members, another possible route was at the school/teacher level through various meetings, particularly for the headteachers. It was quickly known that certain schools were receiving HE treatments, and the control schools were likely to have wished to be treated in the next round of intervention, if happened. Naturally, prior to the project commencement, we had to explain about the HE intervention to the district officer with the presence of several head teachers to obtain an agreement about our field experiment. If regular teachers prompted spillovers, its effect could be partly captured by the stratification variable *school type*, since GPS had higher number of teachers on average; the mean number of teachers for GPS and RNGPS at the baseline were 5.73 (1.48) and 3.72 (0.65), and that of students were 198.0 (75.24) and 126.66 (47.0), respectively (SD in parentheses). A similar thing could be said regarding a female teachers' ratio at baseline that GPS had significantly higher ratio at 58% on average vis-a-vis that of RNGPs at 28%. Note however, that teachers were not the direct target of SBHE and their active role was not readily expected. Below in Table 4, we provide a summary statics of externality measures used for the analysis, namely, *externality index* describe above, and the number of schools *within1 km*, *within 1–2 km*, and *within 2–3 km*.

## Externality model estimation results

We provide externality model estimation results for selected outcomes which had statistically significant HE-treatment effects in the main analysis with CRSE given in Table 3. The externality measures in Table 5 below demonstrates

**Table 4. Summary statistics of externality measures (N = 180).**

| | non-HE school N = 90 | HE Schools N = 90 | |
|---|---|---|---|
| | Mean (SD) | Mean (SD) | p-value[a] |
| HE externality index | 0.44 (0.30) | 0.39 (0.27) | 0.254 |
| number of HE schools within 1 km | 0.33 (0.56) | 0.32 (0.52) | 0.89 |
| number of HE schools within 1–2 km | 1.42 (1.29) | 1.22 (1.10) | 0.264 |
| number of HE schools within 2–3 km | 2.28 (1.50) | 2.17 (1.41) | 0.609 |

Notes: (a) t-test for numerical variables.

significant impacts on (P1) *handwashing*, (P2) *dentalcare*, (P3) *overall hygiene*, (P6) *health/hygiene knowledge*, and (I1) *cold-related symptoms*. Comparing difference externality measures, *externality index* displayed the largest coefficient magnitude, followed by *within 1 km*, *within 1–2 km*, and *within 2–3 km* (omitted here; also estimates for *HE-group*, *school type* and *sex* are omitted). All estimated coefficients showed identical results as those in the original analysis given in Table 3. This not only reaffirms the robustness of *HE*, *period* and other effect estimates, but also indicates that externality effects were at work in addition to *HE-treatment* and *period* effects, and that they improved model's explanatory power. These statistically significant positive spillovers suggest that health-related information and possibly healthier behaviour spread to non-HE school pupils in proximity, with stronger effects amongst those having more HE-schools closer by.

## Cost-effectiveness of health education

The project's direct labour expenses for 18 para-teachers amounted to 14,466 USD for the year. Additionally, there were costs for the initial training workshop and subsequent refresher workshops, totalling 1,156 USD. Beyond these operational expenses, which summed up to 15,622 USD, there were also fixed costs: 4,943 USD for developing digital health education materials and 3,460 USD for purchasing 23 mini-projectors, including five extras. The SBHE digital materials can be used almost perpetually in principle, however, considering that material contents can get outdated, we may conservatively suppose its lifetime as 10 years without major revision. As for the mini-projector, we may assume that such equipment gets depreciated in 5-year time. Presuming their equal use value in each time-period, simply dividing the costs of materials and projector, their annual costs were 494 USD and 346 USD, respectively. Therefore, the total annual cost of SBHE was 16,462 USD including the fixed cost. Per school annual running cost (excluding workshops) was 161 USD or 183 USD including workshops and other fixed cost. Converting the annual cost to monthly cost, it was 13 USD or 15 USD, respectively. This monthly cost of our project was far less than that of the four weekly school hygiene education sessions and hygiene nudge construction project cited above [20] which was 127 USD or 124 USD per school, respectively, which would be 98 USD or 96 USD adjusting for the annual average inflation rate of 6.56% during 2013–2016 in Bangladesh [52]. The number of attended pupils in the HE schools at endline, thus assumed to have been treated during the project, were 15,622 (those in the non-HE schools were 14,381). This makes the per pupil annual cost of SBHE and of para-teachers at 1.05 USD and 0.92 USD, respectively. As a reference, this was twice expensive than per student deworming cost of 0.49 USD in a large scale Tanzanian government project cited in [17].

In terms of cost-effectiveness of HE project, dividing the annual total HE cost per school (183 USD) by the HE-treatment effect on *overall hygiene practices* (0.22 as shown in Table 3 and Table 5) for instance, one SD improvement in child hygiene was achieved by 8.32 USD per school. Divided by the mean attended 173.8 students per HE school at endline, the respective cost-effectiveness was 0.048 USD per pupil. Incorporating the effects of the composite *externality index* shown in Table 5, the total effect of HE become 0.45 SD improvements in *overall hygiene practices*, then one SD improvement in child *overall hygiene practices* in HE schools was achievable by 4.07 USD per school or 0.023 USD

Table 5. Family-wise mean-standardised treatment effect and externality effect on the Child-level selected outcomes.

| | Child-level Mean-standardised Treatment Effects with Externality Effects | | | | | | | | |
|---|---|---|---|---|---|---|---|---|---|
| | (P1) handwashing | | | (P2) dentalcare | | | (P3) overall hygiene | | |
| | externality measures | | | externality measures | | | externality measures | | |
| | externality index | within 1 km | within 1–2 km | externality index | within 1 km | within 1–2 km | externality index | within 1 km | within 1–2 km |
| HE-treatment | 0.21*** | 0.21*** | 0.21*** | 0.17*** | 0.17*** | 0.17*** | 0.22*** | 0.22*** | 0.22*** |
| | [0.13,0.30] | [0.13,0.30] | [0.13,0.30] | [0.08,0.26] | [0.08,0.26] | [0.08,0.26] | [0.14,0.31] | [0.14,0.31] | [0.14,0.31] |
| | [0.000] | [0.000] | [0.000] | [0.000] | [0.000] | [0.000] | [0.000] | [0.000] | [0.000] |
| period | 0.64*** | 0.64*** | 0.64*** | 0.40*** | 0.40*** | 0.40*** | 0.53*** | 0.53*** | 0.53*** |
| | [0.58,0.70] | [0.58,0.70] | [0.58,0.70] | [0.33,0.47] | [0.33,0.47] | [0.33,0.47] | [0.47,0.59] | [0.47,0.59] | [0.47,0.59] |
| | [0.000] | [0.000] | [0.000] | [0.000] | [0.000] | [0.000] | [0.000] | [0.000] | [0.000] |
| externality measures | 0.18** | 0.09*** | 0.04*** | 0.28*** | 0.10** | 0.06*** | 0.23*** | 0.10*** | 0.04*** |
| | [0.06,0.30] | [0.04,0.14] | [0.02,0.06] | [0.16,0.39] | [0.04,0.17] | [0.03,0.08] | [0.12,0.33] | [0.05,0.15] | [0.02,0.06] |
| | [0.003] | [0.000] | [0.000] | [0.000] | [0.002] | [0.000] | [0.000] | [0.000] | [0.000] |
| N | 16181 | 16181 | 16181 | 16183 | 16183 | 16183 | 16163 | 16163 | 16163 |
| | (P6) knowledge | | | (P6E) knowledge+extra (endline) | | | (I1) cold-related symptoms (no CRSE) | | |
| | externality measures | | | externality measures | | | externality measures | | |
| | externality index | within 1 km | within 1–2 km | externality index | within 1 km | within 1–2 km | externality index | within 1 km | within 1–2 km |
| HE-treatment | 0.44*** | 0.44*** | 0.44*** | 0.20*** | 0.20*** | 0.20*** | −0.05** | −0.05** | −0.05** |
| | [0.33,0.55] | [0.33,0.55] | [0.33,0.55] | [0.15,0.25] | [0.15,0.25] | [0.15,0.25] | [-0.08,-0.01] | [-0.08,-0.01] | [-0.08,-0.01] |
| | [0.000] | [0.000] | [0.000] | | | | [0.007] | [0.007] | [0.007] |
| period | 0.80*** | 0.80*** | 0.80*** | | | | −0.05*** | −0.05*** | −0.05*** |
| | [0.72,0.88] | [0.72,0.88] | [0.72,0.88] | | | | [-0.07,-0.02] | [-0.07,-0.02] | [-0.07,-0.02] |
| | [0.000] | [0.000] | [0.000] | | | | [0.000] | [0.000] | [0.000] |
| externality measures | 0.15* | 0.07* | 0.03* | 0.10+ | 0.05+ | 0.02+ | −0.05*** | −0.04*** | 0 |
| | [0.03,0.27] | [0.01,0.13] | [0.01,0.06] | [-0.01,0.21] | [-0.00,0.10] | [-0.00,0.04] | [-0.08,-0.02] | [-0.05,-0.02] | [-0.00,0.01] |
| | [0.018] | [0.021] | [0.016] | [0.089] | [0.056] | [0.081] | [0.001] | [0.000] | [0.526] |
| N | 16183 | 16183 | 16183 | 8991 | 8991 | 8991 | 16183 | 16183 | 16183 |

Notes: Each column represents a separate regression with different externality measures. Outcome families are same as in Table 3 primary outcomes. All estimations apply DID SUR with CRSE, except for (I1) model, and are adjusted for *HE-group*, *school type*, and *sex* (estimates omitted). Externality index incorporate distances from all HE-treatment schools as well as the number of attending pupils ($1/(J \cdot \mu) \cdot \sum_k lnN_k^T \cdot lnN_j \cdot e^{-d_{kj}}$, $1/(J \cdot \mu) \cdot \sum k \ln N k T \cdot \ln N j \cdot e - d k j$, where $d_k$ is the distance of school $j$ from a treatment school $k$, whose effect manifests exponential decay multiplied by the natural logarithm of the total number of attending students in school $j$, $lnN_j$, and that in the treatment school $k$, $lnN_k$, both measured at the baseline, summed up for all $K$ treatment schools, and divided by total number of schools $J$ times μ, the natural logarithm of the average total attending students of all $J$ schools at the baseline, in order to normalise). Other externality measures ("within *x-y* km") correspond to the number of schools within *x-y* km radius. Significance level: + p < 0.1, * p < 0.05, ** p < 0.01, ***p < 0.001; 95% confidence intervals and p-value in brackets. 16183 samples belonged to 90 HE schools and 90 HE-control schools in the baseline and endline.

per pupil, taking into account the spillover effects. The cost-effectiveness of one SD improvement in child-level outcome families are provided in Table 6. As a reference, the above-cited work [18] which conducted community intervention estimated the cost of health knowledge improvement by one percentage to be around 0.75~0.82 USD, and personal hygiene improvement by one percentage to be around 1.10~1.32 USD per household. Although our case estimated the effect size

in terms of an SD unit per child and the content of health education was not the same, thus not directly comparable to previous studies, comparing these HE costs, our school-based SBHE seem to be cost effective.

## Discussions

### Effectiveness of SBHE in rural Bangladesh

The positive outcomes of our SBHE programme in rural Bangladesh affirm its effectiveness in fostering better healthy/hygienic practices and behaviours amongst school children. The evidence also indicates the establishment of new healthier norms, aligning with global findings where school-based health education significantly contributes to it [6,11,21]. A particularly strong evidence of improved handwashing practices through repetition and skill-building corroborates findings from prior research [12,20,53,54]. Enhanced self-hygiene and well-maintained school infrastructure [32] likely serve as preventive measures against infectious diseases, as seen in [11,12,53,55,56]. This was in part confirmed in our analysis, in terms of reduced cold symptoms. The SBHE effects on dentalcare resonated with those of other successful SBHE studies focusing on oral health [6,10,57]. While handwashing and dentalcare habits were primarily based on children's self-reports, the surveyors' evaluations of children's knowledge about proper handwashing techniques lend credibility to these self-reports. Also, the fact that children did not uniformly claim adherence to the recommended practices on all suggested occasions buttressed the objectivity of their responses.

### Limitations and considerations

Nonetheless, the absence of valid ATP measurements indicating the level of hand hygiene at baseline—due to flawed measurement procedures—remains a limitation. Accurate ATP baseline data could have provided a stronger empirical foundation to confirm improvements in hygiene practices. Moreover, the lack of significant impact on nutritional practices and child anthropometry within the study's one-year timeframe suggests that behavioural changes in nutrition are more complex, possibly requiring more integrated and prolonged interventions. This resonates to the findings that neither educational nor food voucher provision alone was effective in improved nutritional practices, but such interventions needed to be provided together [58]. Moreover, as presented in the supplementary S2 Table in S1 Zip, both household wealth and parents' literacy had statistically significant positive effects on nutritional practices, suggesting that financial constraints matter. Regarding child anthropometry, the fact that the growth spurts happen at around puberty might add to the explanation that educational intervention alone was difficult to improve it. Not only on nutrition, but our substantiation that child socioeconomic status positively affected all child healthy practices as well as child health matches the findings in [41]. Given these findings and the fact that there were many malnourished children in terms of international standard, which

**Table 6. Para-teacher and total HE cost in USD for one standard-deviation (SD) improvement in family-wise child health KAPB.**

**Cost per 1 SD improvement in child-level outcome**

| | (P1) handwashing | | (P2) dentalcare | | | (P3) overall hygiene |
|---|---|---|---|---|---|---|
| | para-teacher cost | total cost | para-teacher cost | total cost | para-teacher cost | total cost |
| HE effects | 0.04 | 0.05 | 0.05 | 0.06 | 0.04 | 0.05 |
| HE+spillover | 0.02 | 0.03 | 0.02 | 0.02 | 0.02 | 0.02 |
| | (P6) knowledge | | (P6E) knowledge+extra | | (H2) cold-related symptoms | |
| | para-teacher cost | total cost | para-teacher cost | total cost | para-teacher cost | total cost |
| HE effects | 0.02 | 0.02 | 0.05 | 0.05 | −0.18 | −0.21 |
| HE+spillover | 0.02 | 0.02 | 0.03 | 0.04 | −0.09 | −0.11 |

Notes: Figures shown are calculated running (para-teachers) cost and total cost (including fixed cost) per child in USD per 1SD improvement in outcome families for which estimated coefficient of *HE* and *externality index* had statistical significance of at least 5% given in Table 5.

made us set our anthropometric z-score outlier as –6 SD/+5 SD rather than ±5 SD, an intervention or policy with specific focus on the nutritional enhancement for school children would be recommended. Studies consistently have shown that school feeding programs have a significant positive impact on various aspects of children's education and health, particularly in developing countries [59–61]. Such school feeding policy is thus expected to contribute not only to children's health but also to their learning outcomes in rural Bangladesh. This would also reduce the high incidence of various illness symptoms in children.

Recognising the importance of supply-oriented project for health improvements, there are several plausible reasons which limited our study findings particularly for the secondary outcomes of child health. Unlike interventions providing antihelminth drug [17] or highly intensive interventions aiming for behavioural change with the provision of soap bars and/or hand sanitizers [11,12], a weekly SBHE for one year, encompassing a broad spectrum of health themes, was likely insufficient to make improvement in these outcomes. While our soap treatment provided in a cross-cutting manner was found effective on hygiene practices when provided with the HE-treatment, it had no impact on any of the health outcomes. The cross-cutting intervention design, intended to be cost-effective and efficient in using shared samples to assess two treatments, could have resulted in an insufficient statistical power. Even for the HE-treatment alone, the sample size of 180 schools and the actual effect size might not have provided a sufficient statistical power to detect the effects on health outcomes. Also, the attrition of relatively unhealthy children could have caused a downward bias for the estimated effects. Furthermore, the cancelling-off effects between HE session and PE session, as HE session was carried out using PE class allotment, might have been present if PE classes, playing around, had positive health effects, biasing the estimated HE effects towards zero. Although the inclusion of spillover/externality measures did not affect the effect size of other estimated coefficients, such spillover effects might have diluted the treatment effect estimate itself—often labelled as 'contamination'. These effects, while beneficial for extending the impact of the intervention beyond the immediate target groups, complicate the measurement of direct impacts. Spillover effects were indeed statistically significant; the closer and more numerous the HE-schools, the greater the spillover effect. Additionally, period effects might have also captured part of the positive effects, given the large general improvement from baseline to endline across most outcomes, regardless of the treatment status.

### Reflection on the project implementation and analysis

In terms of the project design and implementation, the study rigorously conducted with randomly chosen samples and treatment allocation, although the soap treatment incurred a few months delay which resulted in irregular distributions. Our SBHE was designed to overcome previously identified difficulties in conducting health education in terms of its cost, context relevance and effectiveness, and unmotivated teachers [18,30,62,63]. Notably, our SBHE employed para-teachers and mobile projectors, which simultaneously overcame the problems of over-burdened primary school teachers, limited budgets, and knowledge gaps among the instructors. The use of mini-projectors with digital materials was also supportive for pupils to acquire healthy behaviours, by engaging them in the context/culturally-relevant and age-relevant animations and images, whose importance has been pointed out by [64].

Analysis-wise, we estimated the mean-standardised average treatment effect on the families of outcomes which made the effects of different factors comparable at the same time avoiding the over identification of statistically significant results. Another strength and the novelty of our study was that we explicitly measured the spillover effects of SBHE. While spillover effects are generally considered as nuisance to the RCT evaluation, in actuality, they could disseminate part of SBHE without extra cost to other schools and pupils. In contrast to the limited spillover effects observed for community-based intervention [65], our school-based intervention demonstrated to be an efficient driver for positive spillovers, especially given the fact that the treatment and control schools were intermingled in the region. Spillover effects could also reduce the de facto cost of intervention, as shown in our cost-effectiveness analysis. The results suggested that relatively uncostly SBHE could induce healthy behaviours not only in treatment schools and their children, but also in other non-treated schools and pupils.

## Implications

Despite limitations, our findings suggest that school-aged children were highly receptive to adopting new health habits, potentially augmented by peer effects within the school environment [27]. Once acquired, healthy habits are non-costly and sustainable per se, and they can reinforce the positive impacts of other supply-based intervention, as proven by the better management and maintenance of school and other hygiene infrastructure [15,32]. To the extent that Jhenaidah district was a typical rural area and that the samples were sufficiently representative of the population, the findings would be generalisable to other rural areas in Bangladesh. Although our study was for a limited period, further research looking into how well such healthy environment and behaviour sustain would be warranted. Recent literature suggesting the effective use of nudging [20,66] is also suggestive of incorporating such tools. Along with economic development, healthy behavioural changes and norms are expected to contribute to breaking the vicious circle of ill health, poor education and poverty, and to overcome the problem of low investment in own health toward sustainable health improvement.

## Conclusions

Our study confirmed that skill-based school health education, emphasising on practical skill-building and context-relevant and age-relevant active learning, is a potent tool for improving health practices among school-aged children in resource-constrained rural settings, with significant positive effects on hygiene behaviour and knowledge. The robustness of these effects, even in the presence of notable spillover, underscores the efficacy of the educational strategies employed. Despite these positive outcomes, the limited impacts on nutritional status and child health point to the need for a more holistic approach that includes dietary interventions. Future research should consider longitudinal designs to capture the long-term effects of SBHE as well as nutritional support, potentially providing a more definitive picture of the intervention's impact on child health and schooling. In conclusion, our research contributes valuable insights into the role of school health education in promoting sustainable health behaviours, with broader implications for public health policy and practice in developing countries. The integration of SBHE into national health and education programs could be a key strategy for achieving long-term health improvements in similar contexts globally.

## Supporting information

**S1 Zip.** **S1 Fig.** Project School Map in Jhenaidah, Bangladesh. **S1 Table.** Endline (non-DID) estimation of family-wise mean-standardised effect in average effect size on nine outcome families adjusting for baseline covariates (all children). **S2 Table.** DID estimation of family-wise mean-standardised effect in average effect size on nine outcome families with additional covariates (all children). **S3 Table.** DID estimation of family-wise mean-standardised cross-cutting HESP-treatment effect in average effect size on five selected outcome families with additional covariates (all children). **S4 Table.** HE-treatment effects on single outcomes (selected outcomes) (all children; children in both surveys) **S1 File.** Study Protocol. **S1 Checklist**. CONSORT Checklist.
(ZIP)

## Acknowledgments

This research project was executed in cooperation with Save the Children, Inc. I am grateful to Shuaib Muhammad and Ziaul Hasan of SURCH in Dhaka, as well as all the recruited project staffs and survey staff for their dedicated work, the officials of the Bangladesh Ministry of Primary and Mass Education and the Director General of Health Services officials, officials of the District of Jhenaidah, as well as the head teachers and all other people who enabled and supported this project in one way or the other. Gratitude extends to all teachers, children and their families for their valuable contributions participating in this project. I also wish to express my deepest condolences for the premature death of Taslima Khatun Gini, one of our field investigators in Jhenaidah, at the time of the endline survey. I am grateful to my colleagues and

various seminar and conference participants for their valuable comments. I am also thankful for the advice provided for field experiment design by Pascaline Dupas, Rachel Glennerster, Marc Shotland, and Iqbal Dhaliwal of Abdul Latif Jameel Poverty Action Lab, MIT. All errors are my own.

## Author contributions

**Conceptualization:** Makiko Omura, Mohini Venkatesh, Ikhtiar Khandaker, Ataur Rahman.

**Data curation:** Makiko Omura.

**Formal analysis:** Makiko Omura.

**Funding acquisition:** Makiko Omura, Mohini Venkatesh.

**Investigation:** Makiko Omura.

**Methodology:** Makiko Omura.

**Project administration:** Makiko Omura, Mohini Venkatesh, Ikhtiar Khandaker, Ataur Rahman.

**Resources:** Makiko Omura.

**Software:** Makiko Omura.

**Supervision:** Makiko Omura.

**Validation:** Makiko Omura.

**Visualization:** Makiko Omura.

**Writing – original draft:** Makiko Omura.

**Writing – review & editing:** Makiko Omura.

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
