## [Decision Letter · Decision Letter 0]

Dear Dr. Omura,

Thank you for submitting your manuscript to PLOS ONE. After careful consideration, we feel that it has merit but does not fully meet PLOS ONE’s publication criteria as it currently stands. Therefore, we invite you to submit a revised version of the manuscript that addresses the points raised during the review process.

We look forward to receiving your revised manuscript.

Kind regards,

Khin Thet Wai, MBBS, MPH, MA

Academic Editor

PLOS ONE

 [This study was enabled by the research grants from the Japan Society for the Promotion of Science — Japanese Grant-in-Aid for Scientific Research (No. 23402033) and the Nomura Foundation, and by in-kind contribution and collaboration with Save the Children Federation, Inc. (originally Save the Children, USA).].  

Additional Editor Comments:

Please revise and modify the manuscript in line with reviewers' comments. Clarity and expansion are deemed necessary concerning with the Introduction, Methods and Discussion Sections.

Reviewers' comments:

Reviewer's Responses to Questions

**Comments to the Author**

1. Is the manuscript technically sound, and do the data support the conclusions?

Reviewer #1: Yes

Reviewer #2: Partly

Reviewer #3: Partly

2. Has the statistical analysis been performed appropriately and rigorously?

Reviewer #1: Yes

Reviewer #2: Yes

Reviewer #3: I Don't Know

3. Have the authors made all data underlying the findings in their manuscript fully available?

Reviewer #1: Yes

Reviewer #2: Yes

Reviewer #3: Yes

4. Is the manuscript presented in an intelligible fashion and written in standard English?

Reviewer #1: Yes

Reviewer #2: No

Reviewer #3: Yes

Reviewer #1: The abstract must be revised as It do not provide information about the Number of School , pupils, teachers and doctors involved . No quantitative data concerning the results of the interventions are shown . Please revise and provide info also about statistical significance of main results observed. Tell us more in the abstract about interventions too. Avoid to repeat "and" and use "as well as

Reviewer #2: Dear Authors,

thank you for submitting your manuscript to PLOS ONE. I am grateful for the opportunity to review your manuscript as part of the journal's publication process.

I found your manuscript titled "The Effects of Skill-Based Health Education - A Randomised-Controlled Intervention in Primary Schools in Rural Banhadesh" very valuable to the field of health education as it presents the results of a comprehensive health education intervention in rural Bangladesh, a rarely studied region. In addition to the extensive RCT, which included 180 randomly selected schools with 7200 pupils at baseline, a vulnerable target group of primary school children was also addressed. It appears that the RCT and all pre-post evaluations were conducted rigorously, with appropriate controls, replications and sample sizes.

However, on the other hand, I found the manuscript quite difficult to understand. In my opinion, the main reason for the lack of clarity could be the structure of the article. In contrast to the implemented SBHE ain primary schools in rural Bangladesh, the manuscript does not appear to be a fully convincing, technically sound scientific report. Therefore, the conclusions do not seem to be sufficiently supported by the data presented. Therefore, I suggest some improvements before publication to make your manuscript stronger and more convincing:

1) The introduction is quite long, unfocused and refers to many different areas of health education in schools (e.g. HIV, seatbelt use). Not all of these areas of health education in schools were analysed as part of the implemented SBHE intervention. The Introduction should focus on the topics that were analysed in the empirical results. Based on the abstract of the manuscript, these are primarily the hygiene status of the school, the health and hygiene practises of the children and their health status. The examination of the variety of different health education in schools and health aspects of children makes the manuscript quite demanding in terms of clarity, rigour and accuracy. Therefore, one may want to consider the possibility of splitting this manuscript into two or more parts and focusing only on hygiene practises in this manuscript. I also suggest splitting the Introduction sectioninto several subsections organised by content.

2) Some details of the description of the methodology are repeated in different subsections. Please revise these carefully and do not repeat the same information in the sections "Project and intervention design", "Samples Selection" and "Results - Project implementation". In my opinion, the "Project implementation" section should be moved from the "Results" section to the "Methods" section.

3) Please try to omit the detailed technical description of the data analysis methods and analysis calculations. Due to the research objective, this is not a methological study and most readers will lose track of the main content if they follow the technical details of the data analysis in detail. In some parts of the Results section and also in the Introduction section, the manuscript gives the impression of a highly technical presentation with details that are not so important for this study, without a common thread and with a lack of clarity. Therefore, these parts need to be rewritten.

4) The Discussion section should be separated from the Conclusion section. For such an extensive project, the results are expected to be discussed in more detail and critically appraised in the context of other comparable studies. The final conclusion of the study should be very clearly formulated and appropriately derived from the data presented. It seems that the empirical part of the manuscript contains many different aspects that challenge the authors to finalise the study in a transparent and convincing way.

5) Please proofread the entire article in English carefully once more. Please define the term "para-teacher" at the point in the manuscript where it is mentioned for the first time.

I hope these suggestions will help you to further develop your manuscript. Thank you for the opportunity to review your article. I wish you good luck with the publication and much success with your valuable project.

Reviewer #3: - The introduction needs restructuring as it is currently disorganized. The introduction is lengthy and should be shortened. Usually, the introduction should not start with the objective of the study. Suggested to a funnel approach: start with a broad statement to set the context and end with the research question/objective. Some references are outdated and may not reflect the current situation.

- How did you decide to select 180 schools?

- Were there any schools unable to complete all 26 modules?

- Were the questionnaires self-administered or conducted as interviews?

- Did you use the same students who participated in the baseline for the endline assessment?

- Did you consider stratification by grades for interviewing the students, or was the sampling done as a whole?

- Some information in the methods is overlapping and repetitive.

- The methods section should explain how the outcome index is measured, not just in a footnote.

- Project implementation and child attrition from results section should be moved to the methods section, including a consort flow diagram.

- For outcomes such as cough, breathing difficulty, fever within 2 weeks, etc., how did you measure them as continuous variables? If these are binary (yes, no), please revise the table as your results are with mean (sd). Otherwise, better explain how you measure these in the methods section, as it can confuse readers.

- Why does the sample size in the analysis starting from Table 4 double? Did you combine baseline and endline sample sizes? If yes, can you explain why you did not just compare the endline results controlling for the baseline results?

- The discussion covers only a few points. There are actually a lot more to be discussed. The discussion section should consist of insights and implications based on findings and comparison with previous studies.

**Do you want your identity to be public for this peer review?** For information about this choice, including consent withdrawal, please see our Privacy Policy

Reviewer #1: No

Reviewer #2: No

Reviewer #3: No

---

## [Author Response · Author response to Decision Letter 1]

4 Oct 2024

Response to the Reviewers

We are grateful for the insightful and detailed feedback from all reviewers. We are extremely grateful for the insightful and detailed feedback from all reviewers. Your suggestions have significantly improved the manuscript, and we have made every effort to address your concerns as described below in blue.

Reviewer #1: The abstract must be revised as It do not provide information about the Number of School , pupils, teachers and doctors involved . No quantitative data concerning the results of the interventions are shown . Please revise and provide info also about statistical significance of main results observed. Tell us more in the abstract about interventions too. Avoid to repeat "and" and use "as well as

*Response:

The abstract has been revised to include the information of the number of schools and pupils who were the target of intervention as well as the subjects of survey. Quantitative data on the main analysis results. are added. Repeated “and” is replaced with “as well as”.

Reviewer #2: Dear Authors,

thank you for submitting your manuscript to PLOS ONE. I am grateful for the opportunity to review your manuscript as part of the journal's publication process.

I found your manuscript titled "The Effects of Skill-Based Health Education - A Randomised-Controlled Intervention in Primary Schools in Rural Bangladesh" very valuable to the field of health education as it presents the results of a comprehensive health education intervention in rural Bangladesh, a rarely studied region. In addition to the extensive RCT, which included 180 randomly selected schools with 7200 pupils at baseline, a vulnerable target group of primary school children was also addressed. It appears that the RCT and all pre-post evaluations were conducted rigorously, with appropriate controls, replications and sample sizes.

However, on the other hand, I found the manuscript quite difficult to understand. In my opinion, the main reason for the lack of clarity could be the structure of the article. In contrast to the implemented SBHE in primary schools in rural Bangladesh, the manuscript does not appear to be a fully convincing, technically sound scientific report. Therefore, the conclusions do not seem to be sufficiently supported by the data presented. Therefore, I suggest some improvements before publication to make your manuscript stronger and more convincing:

1) The introduction is quite long, unfocused and refers to many different areas of health education in schools (e.g. HIV, seatbelt use). Not all of these areas of health education in schools were analysed as part of the implemented SBHE intervention. The Introduction should focus on the topics that were analysed in the empirical results. Based on the abstract of the manuscript, these are primarily the hygiene status of the school, the health and hygiene practises of the children and their health status. The examination of the variety of different health education in schools and health aspects of children makes the manuscript quite demanding in terms of clarity, rigour and accuracy. Therefore, one may want to consider the possibility of splitting this manuscript into two or more parts and focusing only on hygiene practises in this manuscript. I also suggest splitting the Introduction section into several subsections organised by content.

*Response:

Given relatively few RTC of health education intervention, references were made to other areas of health education. However, in order not to lose the focus, as recommended, the introduction has been streamlined. We included various measured outcomes in this paper, given Plos One policy not to split the manuscript. However, given the complexity of contents as pointed out, we decide to focus only on the child-level analysis in this manuscript. Given the initial project objective to improve both hygiene status of school and children, as well as health status of children, it was considered to better to split the paper by target and data levels. This also allows us to present the positive findings in behaviour change and yet the difficulty of finding health impacts in a relatively short-term.

What has been done:

• Manuscript is split to present only child level analysis.

• Introduction section is considerably shortened and restructured for clarity with added subsections.

2) Some details of the description of the methodology are repeated in different subsections. Please revise these carefully and do not repeat the same information in the sections "Project and intervention design", "Samples Selection" and "Results - Project implementation". In my opinion, the "Project implementation" section should be moved from the "Results" section to the "Methods" section.

*Response:

What has been done:

• Methods section is restructured with additional subsection headings for clarity and not to be repetitive.

• "Attrition" and "Project implementation" are moved to Method section.

3) Please try to omit the detailed technical description of the data analysis methods and analysis calculations. Due to the research objective, this is not a methological study and most readers will lose track of the main content if they follow the technical details of the data analysis in detail. In some parts of the Results section and also in the Introduction section, the manuscript gives the impression of a highly technical presentation with details that are not so important for this study, without a common thread and with a lack of clarity. Therefore, these parts need to be rewritten.

*Response:

While we appreciate the concerns regarding potential distractions from excessive technicalities, it is customary to present technical details in the field of economics to ensure the rigour of analysis. Since Plos One does not permit footnotes, and to accommodate diverse readerships, we have streamlined some technical contents, and added a subsection "Treatment of intracluster correlation" to contain further technical details that are deemed necessary for the justification of rigorous analysis. In this way, non-interested readers can skip this subsection which may distract them from the main study thread, while still catering to those who value detailed methodological insights.

4) The Discussion section should be separated from the Conclusion section. For such an extensive project, the results are expected to be discussed in more detail and critically appraised in the context of other comparable studies. The final conclusion of the study should be very clearly formulated and appropriately derived from the data presented. It seems that the empirical part of the manuscript contains many different aspects that challenge the authors to finalise the study in a transparent and convincing way.

*Response:

What has been done:

• Discussion and Conclusion sections are separately presented; Discussion is separated into four subsections for clarity.

• In the Discussion section, we discussed the results more extensively and provided insights and implications in the context of other comparable studies, covering all child-level outcomes, i.e., hygiene practices, nutritional practices, and health outcomes.

• Conclusions succinctly summarise the study findings, and propose possible future research directions.

5) Please proofread the entire article in English carefully once more. Please define the term "para-teacher" at the point in the manuscript where it is mentioned for the first time.

*Response:

What has been done:

• Proofread the article and revised it

• the first appearance of "para-teacher" in Introduction changed to a generic term "instructor"; para-teacher hence defined in the project context at its first appearance

I hope these suggestions will help you to further develop your manuscript. Thank you for the opportunity to review your article. I wish you good luck with the publication and much success with your valuable project.

Reviewer #3: - The introduction needs restructuring as it is currently disorganized. The introduction is lengthy and should be shortened. Usually, the introduction should not start with the objective of the study. Suggested to a funnel approach: start with a broad statement to set the context and end with the research question/objective. Some references are outdated and may not reflect the current situation.

*→ the introduction restructured, shortened and revised with additional subsections

- How did you decide to select 180 schools?

*→ information added: due to budget and logistical constraints ("Randomisation and intervention description" subsection)

- Were there any schools unable to complete all 26 modules?

*→ all schools completed 26 modules with some additional repetition; description corrected as modules and topics were mixed up ("Project Implementation" subsection).

- Were the questionnaires self-administered or conducted as interviews?

*→ as noted in the Methods section (with added subsection heading "Data collection"), that surveyors conducted the interviews

- Did you use the same students who participated in the baseline for the endline assessment?

*→ Yes those baseline students were followed up at endline, with replacement students for those attritted students; additional statement added for clarity ("Data collection" subsection)

- Did you consider stratification by grades for interviewing the students, or was the sampling done as a whole?

*→ Thank you - that information was missing; the information added ("Sample size calculation " subsection); the sampling of equal number (10) was done per grade/class

- Some information in the methods is overlapping and repetitive.

*→ the method section was restructure and amended to avoid repetitions

- The methods section should explain how the outcome index is measured, not just in a footnote.

*→ information added in the Method section, Outcome subsection

- Project implementation and child attrition from results section should be moved to the methods section, including a consort flow diagram.

*→ "Project implementation" and "Child attrition" moved from Result section to Method section, as well as the Consort flow diagram

- For outcomes such as cough, breathing difficulty, fever within 2 weeks, etc., how did you measure them as continuous variables? If these are binary (yes, no), please revise the table as your results are with mean (sd). Otherwise, better explain how you measure these in the methods section, as it can confuse readers.

*→ They are measured as yes/no, however, in the family-wise mean-standardised treatment effect estimation model, all binary variables {0,1} are treated as continuous as it essentially calculates the probability of outcome occurrence (as noted in the Statistical Analysis subsection of the Method section). Additional note is given in the Methods section as well as in Table 2 to avoid confusion. Additionally, although logistic regressions are run for each variable analysis presented in the Supplement, binary variables mean values provides a direct indication of the proportion of positive responses. Since they can be straightforwardly transformed into frequency measures, it is omitted to save the length of the already lengthy table.

- Why does the sample size in the analysis starting from Table 4 double? Did you combine baseline and endline sample sizes? If yes, can you explain why you did not just compare the endline results controlling for the baseline results?

*→ Since the analysis utilises both baseline and endline data (including the outcome variables) for difference-in-differences (DID) estimation as well as other types of estimations, the sample size becomes double. As noted in the Statistical Analysis subsection, the application of DID adjusts for any pre-existing differences between treatment and comparison schools and enhances the statistical power and precision of our findings. By incorporating both time points, we ensure that observed effects are attributable to the intervention, isolating it from other temporal trends.

However, we added the non-DID estimation results in S1 Table, which provide the intent-to-treat (ITT) effects differences between treatment and control groups in terms of family-wise mean-standardised average treatment effect, controlling for all covariates and the baseline outcome values. Both the DID and non-DID models are estimated applying the seemingly unrelated regressions (SUR) for a family of grouped outcomes, and all outcomes are normalised by the control group baseline mean and standard deviation, as specified in Kling et al (Econometrica 2007) and Clingingsmith et al (Quarterly J. of Economics 2009).

- The discussion covers only a few points. There are actually a lot more to be discussed. The discussion section should consist of insights and implications based on findings and comparison with previous studies.

*→ We separated Discussion and Conclusion sections, and the Discussion section now contains four subsections for clarity. Results are discussed more extensively, providing insights and implications also in the context of other comparable studies, covering all child-level outcomes: hygiene practices; nutritional practices; health outcomes. Conclusions succinctly summarise the study findings, and propose possible future research directions.

---

## [Decision Letter · Decision Letter 1]

The Effects of Skill-Based Health Education --- A Randomised-Controlled Intervention in Primary Schools in Rural Bangladesh

PONE-D-24-17220R1

Dear Dr. Omura,

We’re pleased to inform you that your manuscript has been judged scientifically suitable for publication and will be formally accepted for publication once it meets all outstanding technical requirements.

Kind regards,

Khin Thet Wai, MBBS, MPH, MA

Academic Editor

PLOS ONE

Additional Editor Comments (optional):

Reviewers' comments:

Reviewer's Responses to Questions

**Comments to the Author**

Reviewer #3: All comments have been addressed

2. Is the manuscript technically sound, and do the data support the conclusions?

Reviewer #3: Yes

3. Has the statistical analysis been performed appropriately and rigorously?

Reviewer #3: Yes

4. Have the authors made all data underlying the findings in their manuscript fully available?

Reviewer #3: Yes

5. Is the manuscript presented in an intelligible fashion and written in standard English?

Reviewer #3: Yes

Reviewer #3: (No Response)

**Do you want your identity to be public for this peer review?** For information about this choice, including consent withdrawal, please see our Privacy Policy

Reviewer #3: No

---

## [Editor Report · Acceptance letter]

PONE-D-24-17220R1

PLOS ONE

Dear Dr. Omura,

I'm pleased to inform you that your manuscript has been deemed suitable for publication in PLOS ONE. Congratulations! Your manuscript is now being handed over to our production team.

Kind regards,

on behalf of

Dr. Khin Thet Wai

Academic Editor

PLOS ONE